# `zip2zip`: Inference-Time Adaptive Tokenization via Online Compression

**Saibo Geng**[1*]   **Nathan Ranchin**[1*]   **Yunzhen Yao**[1]   **Maxime Peyrard**[4]
**Chris Wendler**[1,2]   **Michael Gastpar**[1]   **Robert West**[1]
[1]EPFL   [2]Northeastern University
[4]Université Grenoble Alpes, CNRS, Grenoble INP, LIG
{saibo.geng, nathan.ranchin, yunzhen.yao, michael.gastpar, robert.west}@epfl.ch
maxime.peyrard@univ-grenoble-alpes.fr   ch.wendler@northeastern.edu

## Abstract

Tokenization efficiency plays a critical role in the performance and cost of large language models (LLMs), yet most models rely on static tokenizers optimized on general-purpose corpora. These tokenizers' fixed vocabularies often fail to adapt to domain- or language-specific inputs, leading to longer token sequences and higher computational costs. We introduce `zip2zip`, a novel method for achieving context-adaptive tokenization in LLMs at inference time. Leveraging an online data compression algorithm (Lempel–Ziv–Welch), zip2zip dynamically expands its active vocabulary at inference time by continuously replacing fragmented token sequences with more compact hypertokens, which it can immediately output during generation. `zip2zip` consists of three key components: (1) a tokenizer based on Lempel–Ziv–Welch compression that incrementally merges co-occurring tokens into reusable hypertokens on the fly; (2) a dynamic embedding (and unembedding) layer that computes embeddings for newly formed hypertokens at runtime; and (3) a variant of autoregressive language modeling that pretrains the model to handle hypertokenized, compressed text sequences as inputs and outputs. We show that an existing LLM can be uptrained for zip2zip in 10 GPU-hours via parameter-efficient finetuning. The resulting LLM performs test-time adaptation, learning to use hypertokens in unseen contexts and reducing input and output tokens by 15–40%. Code and models are released at https://github.com/epfl-dlab/zip2zip.

## 1   Introduction

Large language models (LLMs) have shown impressive versatility across a broad spectrum of tasks and domains [Brown et al., 2020, Bubeck et al., 2023], including biomedical tests [Nori et al., 2023], mathematical reasoning [Frieder et al., 2023], programming [Jiang et al., 2024], and multiple human languages. A critical underlying component of this flexibility is the tokenizer, which defines the model's vocabulary and governs how raw text is converted into token sequences fed to the model. The efficiency of the tokenization scheme—i.e., how compactly a text is represented as tokens—has significant impact on model performance. In particular, a more compact tokenization yields three key benefits: (1) larger effective context windows; (2) lower computational (and thus monetary) cost; and (3) shorter response times.

Despite its importance, the tokenizers used in most LLMs operate with fixed, static vocabularies obtained by running algorithms such as Byte Pair Encoding [Sennrich et al., 2016] over large-scale, general-purpose web corpora. While this globally optimized vocabulary performs reasonably well on

---

[*]Equal contribution.

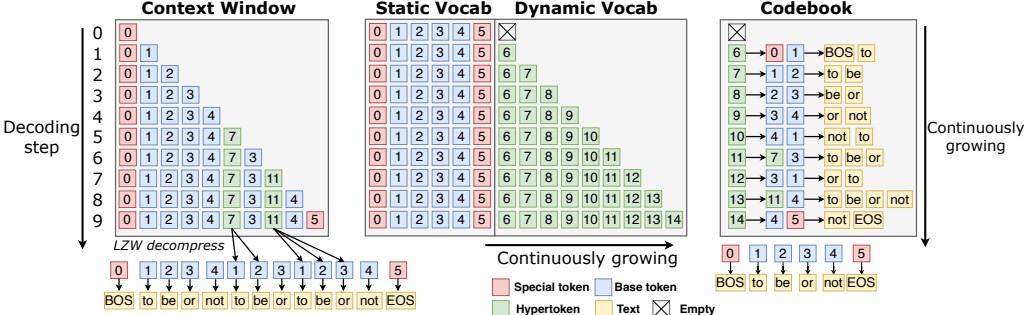

Figure 1: **zip2zip inference process.** At each decoding step, the model has a growing **context** composed of both **base tokens** (blue) and **hypertokens** (green). The **static vocabulary** of size 6 remains fixed, while the **dynamic vocabulary** is continuously expanded by merging co-occurring tokens using **LZW compression.** The **codebook** (right) maps hypertoken IDs to their corresponding base tokens. As decoding progresses, new hypertokens created at step $t$ (e.g., "to be", "or not") become immediately available for reuse at step $t + 1$. Hypertokens are also eligible for merging, enabling the formation of **nested hypertokens.** The final output sequence (bottom) is reconstructed via LZW decompression.

average, it often fails to adapt to domain-specific or language-specific distributions [Ahia et al., 2023, Petrov et al., 2023], where the text distribution diverges significantly from the pretraining data. The resulting mismatch leads to longer token sequences, increasing both memory and compute demands, as well as the end user's cost by a factor of 2–3x when processing domain-specific text [Ahia et al., 2023]. To mitigate this issue, prior work has explored expanding the token vocabulary during domain or language adaptation to improve tokenization efficiency [Wang et al., 2019, Zhao et al., 2024, Kim et al., 2024, Liu et al., 2023, 2024a]. While effective, this approach needs to be repeated for each target domain or language and requires maintaining separate tokenizers. Meanwhile, commercial LLM providers trend toward increasing the size of token vocabularies—growing from 32K to 128K [Grattafiori et al., 2024] and even up to 200K [Abdin et al., 2024] tokens—to improve overall tokenization efficiency. However, prior work [Dagan et al., 2024, Liang et al., 2023] shows that simply enlarging the vocabulary yields diminishing returns in domain adaptation, and vocabularies past a certain size can potentially degrade model performance [Liang et al., 2023]. These limitations point to a compelling need for an adaptive tokenization mechanism—one that can dynamically tailor the vocabulary to the input text at inference time, without retraining the model or maintaining separate tokenizers. Such a mechanism would allow the model to construct new domain-specific tokens on-the-fly, so as to enhance tokenization efficiency. However, adaptive tokenization poses architectural challenges, as both the embedding layer and the language modeling head in transformer models [Vaswani et al., 2017] are static matrices tied to a fixed vocabulary size.

In this paper, we propose zip2zip (with a hat-tip to seq2seq [Sutskever et al., 2014]), a novel building block that brings inference-time adaptive tokenization to LLMs. zip2zip comprises three key components: (1) **LZW tokenizer:** A tokenizer that integrates the Lempel–Ziv–Welch[2] compression algorithm on top of Byte Pair Encoding (BPE) [Welch, 1984]. By applying the LZW compression algorithm to the base token sequence—continuously merging frequently co-occurring token sequences into reusable longer tokens (hypertokens)—the resulting tokenization becomes less fragmented and more compact. (2) **Dynamic-embedding architecture:** An augmentation of the transformer architecture with a lightweight encoder that replaces the static embedding matrix, allowing the model to compute embeddings for newly formed hypertokens on the fly. (3) **Pretraining under online token compression:** a variant of causal language modeling that trains the model directly on LZW-compressed sequences, aligning learning with the inference-time (hyper)token distribution. The overall process is illustrated in Figure 1, which shows how the context window, dynamic vocabulary, and codebook evolve together during decoding. The name zip2zip reflects its dual role in achieving compression of both the input tokens (the first *zip*) and output tokens (the second *zip*), thereby jointly improving the efficiency of input encoding and output decoding. We conduct continued pretraining on Phi-3-4B and Phi-3.5-14B to support zip2zip using as few as 100M tokens. The resulting models demonstrate strong inference-time compression capabilities across various domains, achieving 15–40% reductions in sequence length and up to 40% improvements in end-to-end latency.

---

[2]LZW is the algorithm used in the ZIP compression tool, which inspired the name zip2zip.

To make it easy to upgrade existing LLMs to zip2zip, we release an efficient, open-source implementation of the training and inference stack. It includes (1) a fast Rust-based LZW tokenizer, (2) a drop-in model architecture compatible with HuggingFace Transformers, (3) a training pipeline for LZW-compression-based finetuning. Existing LLMs can be seamlessly extended with zip2zip, gaining adaptive tokenization capabilities through parameter-efficient finetuning.

## 2 zip2zip

### 2.1 Dynamic Token Vocabulary

To enable dynamic tokenization at inference time, we associate the LLM with a *hyper-vocabulary* $\mathcal{V}_h$ that augments the model's static token vocabulary. Tokens from the original vocabulary $\mathcal{V}$ are referred to as *base tokens*. Each entry in the hyper-vocabulary is a *hypertoken*, representing a merged sequence of base tokens. The total vocabulary for a zip2zip model is the union $\mathcal{V} \cup \mathcal{V}_h$. At the beginning of each inference session, $\mathcal{V}_h$ is initialized as an empty set, and is incrementally populated during decoding by identifying and merging recurring token subsequences in the context window, as illustrated in Figure 1.

**Continuous Vocabulary Expansion.** As decoding proceeds, zip2zip continuously merges co-occurring tokens into new hypertokens and recursively applies merging on new hypertokens. This *continual expansion* allows the model to represent longer, recurring sequences of base tokens compactly. Hypertokens are treated as first-class tokens within the model, used interchangeably with base tokens throughout the decoding process. Importantly, this process occurs entirely during inference, without modifying the underlying tokenizer or requiring model retraining.

**LZW Algorithm.** We implement vocabulary expansion using the Lempel–Ziv–Welch (LZW) compression algorithm—a dictionary-based, lossless compression method that incrementally builds a codebook of variable-length sequences. In our setting, the codebook is initialized with the base token vocabulary $\mathcal{V}$ and expands by adding new hypertokens on the fly as recurring token patterns are encountered. To control the growth of the dynamically expanding vocabulary, we impose a maximum merge size $M$ that restricts how many base tokens a single hypertoken can represent. LZW is particularly well-suited for zip2zip due to the following properties:

(1) it is **online:** hypertokens created at step $t$ can be immediately reusable at step $t+1$; in contrast, methods like BPE require access to the full sequence and operate offline;

(2) it is **self-contained:** input base tokens can be perfectly reconstructed from the compressed token sequence alone;[3]

(3) it is **unambiguous:** when both base tokens and hypertokens are available, which one to use is consistently determined by the LZW algorithm without ambiguity.

### 2.2 Hyper-Embedding and Hyper-Unembedding

Hypertokens do not have fixed embedding vectors in the original model's embedding layer (and unembedding layer), as they are not part of the original vocabulary. To compute the embedding of a hypertoken, we learn a mapping from the base token embeddings to the hypertoken embedding. We achieve this by introducing a *hyper-encoder*, which is a neural network that takes the embeddings of the constituent base tokens as input and outputs the corresponding hypertoken embedding (see Figure 2(a)). Specifically, for a sequence of $M$ base tokens $y_{1:M} := y_1 \ldots y_M$, the hyper-encoder $f_\phi : \mathcal{V}^M \to \mathbb{R}^d$ produces the hypertoken embedding $h = f_\phi(y_{1:M}) \in \mathbb{R}^d$, where $M$ is the maximum merge size and $d$ is the embedding dimension. For hypertokens composed of fewer than $M$ base tokens, we pad the input sequence to length $M$. Since the embedding map for base tokens remains unchanged, the hyper-encoder $f_\phi$ essentially maps the concatenated base token embeddings from an $(M \times d)$-dimensional space to a $d$-dimensional hypertoken embedding vector, performing nonlinear dimensionality reduction. For the output unembedding layer, if the underlying transformer ties the embedding and the unembedding matrices, one can reuse the same hyper-encoder to compute the representation used for unembedding. Otherwise, a separate hyper-encoder is trained to produce the hypertoken unembedding vectors.

---

[3]There is no need to persist or transmit the codebook across inference calls, preserving compatibility with existing LLM libraries and interfaces.

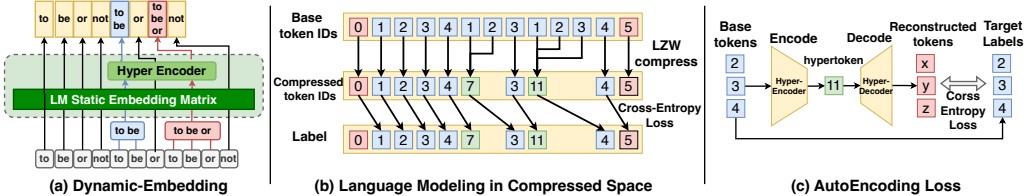

**(a) Dynamic-Embedding**   **(b) Language Modeling in Compressed Space**   **(c) AutoEncoding Loss**

Figure 2: (a) **Dynamic embedding**: Base tokens are embedded via a static LM embedding matrix, while hypertokens (e.g., "to be" or "to be or") are dynamically composed using a hyper-encoder over their constituent base tokens. (b) **Language modeling in compressed space**: The model is trained to predict compressed token sequences produced by LZW, optimizing cross-entropy loss over compressed token IDs. (c) **Auto-encoding loss**: To ensure hypertokens are semantically consistent with their base-token compositions, the model also learns to reconstruct the original base tokens from the hyper-token via a decoding loss.

## 2.3 Architecture

We illustrate the `zip2zip` architecture in Figure 3. The input text is first tokenized into base tokens (**STEP 1**), which are then passed through an online LZW compressing module that compresses the token sequence into a stream of hypertokens (**STEP 2**). Since hypertokens are not part of the model's original embedding layer, their embedding vectors are computed on-the-fly using the *hyper-encoder* during inference (**STEP 3–4**). Once embedded, both base token embeddings and hypertokens embeddings are passed through the standard transformer layers of the base model, producing contextualized hidden states (**STEP 5–6**). This step is identical to vanilla transformer, with hypertokens and base tokens treated equally. At the output unembedding layer, hypertoken unembedding vectors (same as the hypertoken embedding vectors in the tied case, and computed by a separate hyper-encoder otherwise) are appended to the original unembedding matrix in the language modeling head (**STEP 7**). This allows the model to compute a joint softmax over the union of the base vocabulary and the hyper vocabulary $\mathcal{V} \cup \mathcal{V}_h$ (**STEP 8**). The resulting probability distribution

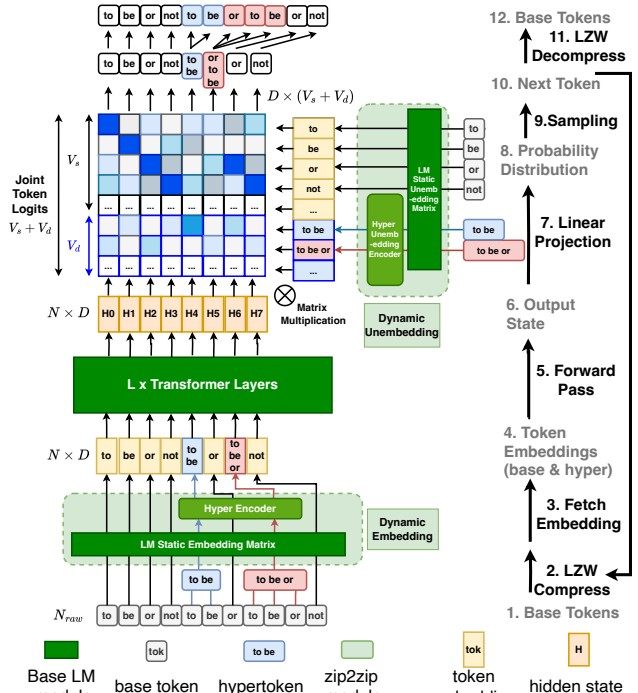

Figure 3: **`zip2zip` architecture and pipeline.** At inference time, base tokens are compressed into hypertokens using LZW. A hyper-encoder computes embeddings for hypertokens, which are processed by the base LLM. Output representations are projected jointly on base and hyper-unembedding layers, producing joint logits and sampled tokens, which can be decoded back to base tokens.

is over $\mathcal{V} \cup \mathcal{V}_h$, and the sampled token may be either a base token or a hypertoken (**STEP 9**). In the next cycle, the newly generated token (**STEP 10**)—whether base or hyper—is appended to the input sequence, and the process repeats (back to **STEP 1**). At the end of generation, the hypertoken sequence is decompressed via the LZW decoding function into a sequence of base tokens (**STEP 11–12**). The whole process works in a fully *autoregressive* way, where newly generated hypertokens will also be merged into new hypertokens for future steps. Furthermore, we highlight two points:

**Consistent Vocabulary Updates.** The expanding vocabulary—comprising newly created hypertokens—must be updated in a *consistent* manner across both the input embedding layer and the output unembedding layer, maintaining a consistent view of the hypertoken set. Failure to update both sides

consistently can result in two types of errors: (1) hypertokens that cannot be decoded, or (2) the model attempting to decode a non-existing hypertoken.

**Hyper-Embedding Cache.** Although hypertoken embeddings are computed on-the-fly, they are context-independent and can thus be cached across inference steps. Similar to the transformer's KV-cache, this enables *incremental* updates: only newly created hypertokens need to be embedded at each step. Since the codebook grows linearly with the number of tokens in the context, the total cache size also grows linearly in memory. Thus, the computational cost for hypertoken embeddings remains constant per step—i.e., one token embedding is computed per step.

## 2.4 `zip2zip` Pretraining

**Objective.** Let $\mathcal{D}$ denote the target text distribution. Given a language model $\pi_\theta$ parameterized by $\theta$, standard pretraining seeks to minimize the causal language modeling (CLM) objective (see Figure 2(b)), which corresponds to the expected negative log-probability of data sequences under the model:

$$\min_\theta \mathbb{E}_{y\sim\mathcal{D}}\left[-\log\pi_\theta(y)\right], \tag{1}$$

where $\pi_\theta(y)$ denotes the probability of the token sequence $y$ under the model $\pi_\theta$.

Let $\mathcal{C}$ be an *online* compression algorithm (e.g., LZW), and $\phi$ be the parameters of the hyper-encoder. Given a sequence $y \sim \mathcal{D}$, let $z = \mathcal{C}(y)$ be its compressed form. In `zip2zip`, we aim to optimize the same CLM loss, but over the compressed sequences $z$. The training objective becomes:

$$\min_{\theta,\phi} \mathbb{E}_{y\sim\mathcal{D}}\left[-\log\pi_{\theta,\phi}(\mathcal{C}(y))\right] = \min_{\theta,\phi} \mathbb{E}_{z\sim\mathcal{C}(\mathcal{D})}\left[-\log\pi_{\theta,\phi}(z)\right]. \tag{2}$$

Here, we slightly abuse the notation to let $\pi_{\theta,\phi}(z)$ denote the probability assigned to the compressed sequence $z$, parameterized by the base model weights $\theta$ and the hyper-encoder parameters $\phi$.

To construct the compressed dataset $\mathcal{C}(\mathcal{D})$, we first tokenize the corpus using a standard tokenizer, and then apply the LZW compression algorithm. This preprocessing step is performed once prior to training and can be efficiently parallelized through batching. Compression is applied at the document level, meaning that each document is compressed independently. This prevents the compressor from learning patterns across unrelated documents.

**Parallelizable Training via Causal Masking.** Although hypertokens introduce additional vocabulary dynamics, training remains fully parallelizable. We leverage the standard causal masking mechanism used in language models, allowing the model to predict the next token—whether a base token or a hypertoken—at each position in parallel. To eliminate the need for sequential codebook updates during inference, we precompute a fixed codebook by applying LZW compression to the entire input sequence. This precomputed codebook is then used consistently throughout training to condition token predictions, ensuring efficiency and compatibility with standard training pipelines.

**Auxiliary Reconstruction Loss.** We introduce an auxiliary reconstruction objective that encourages a hypertoken embedding to retain sufficient information about its underlying base token sequence (see Figure 2(c)). Specifically, the model is trained to reconstruct the original base token embeddings from the hypertoken embedding. We jointly optimize the language model and the hyper-encoder using a combined loss that includes both the standard next-token prediction loss and the auxiliary reconstruction loss. Formally, we optimize:

$$\min_{\theta,\phi,\psi} \mathbb{E}_{y\sim\mathcal{D}}\left[-\log\pi_{\theta,\phi}(\mathcal{C}(y))\right] + \lambda\,\mathbb{E}_{y_{1:M}}\left[\Delta\left(y_{1:M}, f_\psi\left(f_\phi(y_{1:M})\right)\right)\right], \tag{3}$$

where $f_\phi : \mathcal{V}^M \to \mathbb{R}^d$ is the hyper-encoder, $f_\psi : \mathbb{R}^d \to \mathcal{V}^M$ is the decoder aiming to reconstruct the corresponding base tokens from their hyper-embedding, and $\Delta : \mathcal{V}^M \times \mathcal{V}^M \to \mathbb{R}$ is the reconstruction loss function, such as the cross-entropy loss, between the base tokens $y_{1:M}$ and the reconstructed base tokens $f_\psi\left(f_\phi(y_{1:M})\right)$. The hyperparameter $\lambda \geq 0$ controls the trade-off between the prediction error of the language model and the reconstruction error of the autoencoder. This joint optimization objective encourages the hyper-encoder to learn a compact $d$-dimensional manifold embedded in the higher-dimensional $(M \times d)$ space of base token embeddings, while the language model $\pi_{\theta,\phi}$ learns to predict the next (hyper)token given the preceding context. The reconstruction loss can be viewed as a form of auto-encoding, where the hypertoken acts as a compressed latent representation and reconstruction encourages the preservation of semantic content and the compression to be lossless.

**Adapting Pretrained Language Models.** Retraining large language models from scratch is computationally expensive and often infeasible for most research labs. A more economical alternative is to perform continued pretraining (or adaptation) on existing pretrained model weights. The proposed objectives (Equations 2, 3) integrate naturally into this setup. Parameter-efficient methods such as LoRA [Hu et al., 2022] may also be used, which allow selectively updating parts of the base model weights with minimal computational cost.

## 2.5 Efficiency Advantage

`zip2zip` improves efficiency by increasing the average token length, thereby reducing the number of tokens required to represent the same text. This compression applies to both inputs (e.g., prompts) and outputs (e.g., completions). As a result, the model performs fewer computations—both in the attention mechanism and the feedforward layers—and, more importantly, requires fewer autoregressive decoding steps during inference. Since the latency of large language models is primarily driven by the cost of sequential decoding, reducing the number of output tokens by $n\%$ leads to an approximate $n\%$ speedup in decoding latency, which we will demonstrate empirically in Section 3.6. A more detailed discussion of FLOPs is provided in Appendix E for completeness.

## 2.6 Entropy Invariance under Lossless Compression

Before turning to empirical results, we analyze whether a lossless compression of the data representation can fundamentally alter the achievable performance of a model. We show that for any lossless mapping $g$, there always exists a corresponding *transported model* distribution in the compressed space that achieves exactly the same (cross-)entropy as in the original space.

Let $\mathcal{X}$ be the original alphabet and $\mathcal{Z}$ an arbitrary alphabet obtained via a lossless compressor $g : \mathcal{X}^* \to \mathcal{Z}^*$, which is a bijection onto its image. Denote by $P_X$ the true distribution over sequences $x \in \mathcal{X}^*$, and by $p_\theta$ the model distribution on the same space. The corresponding push-forward (compressed-space) true distribution $P_Z$ and model distribution $p_\gamma$ are defined as

$$P_Z(z) = P_X(g^{-1}(z)), \qquad p_\gamma(z) = p_\theta(g^{-1}(z)), \qquad z \in \mathcal{Z}^*.$$

**Theorem 2.1** (Entropy invariance under lossless compression)**.** *If $g$ is lossless (i.e., bijective onto its image), then the total entropy and cross-entropy are invariant under the transformation:*

$$H(P_Z) = H(P_X), \qquad H(P_Z, p_\gamma) = H(P_X, p_\theta).$$

A detailed proof is provided in Appendix G.

The theorem implies that the optimal achievable cross-entropy in the compressed representation is identical to that in the original domain: for any model family on $\mathcal{X}^*$, one can always construct a corresponding model family on $\mathcal{Z}^*$ via push-forward that attains the same likelihood. In practice, the training process can be viewed as an attempt to approximate this *transported model* through optimization; however, convergence to the target model is not guaranteed (see Section 5).

# 3 Experiments

To evaluate the effectiveness of `zip2zip`, we perform continued pretraining on the Phi-3 models (3B and 14B) within the `zip2zip` framework. We train a single model on a general-purpose corpus and evaluate it across four dimensions: (1) token efficiency, (2) language modeling perplexity, (3) downstream task performance, and (4) inference efficiency. This setup allows us to assess how well `zip2zip` generalizes to diverse domains without any task- or domain-specific fine-tuning. For perplexity and downstream benchmarks, we use the widely adopted lm-evaluation-harness framework [Gao et al., 2024].

## 3.1 Training Setup

Rather than updating the full model weights, we adopt parameter-efficient finetuning using LoRA [Hu et al., 2022]. In addition, we train the *hyper-embedding* and *hyper-unembedding* modules. We set the maximum merge size to $M = 3$ and use a two-layer transformer encoder as the hyper-encoder.

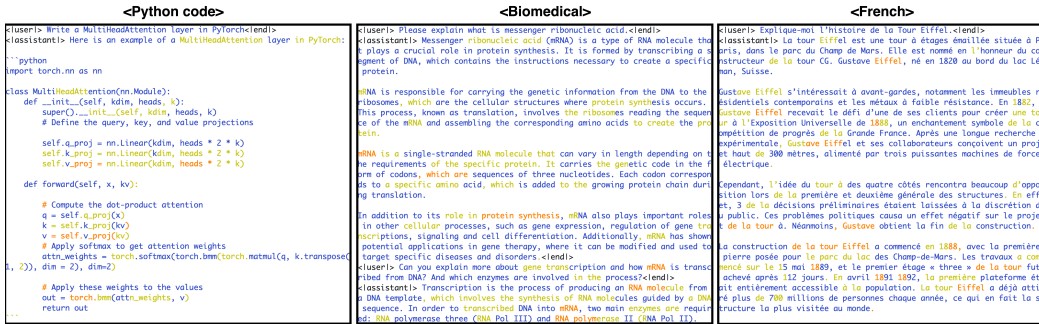

Figure 4: **Phi-3.5-zip2zip output examples.** Blue: base tokens. Yellow: hypertokens (composed of 2 base tokens). Orange: hypertokens (composed of 3+ base tokens).

Table 1: Examples of hypertokens formed by Phi-3.5-zip2zip across three domains

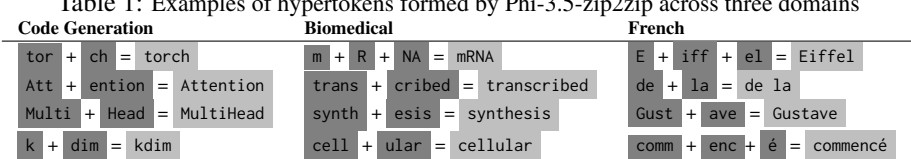

The loss weighting coefficient $\lambda$ was chosen to be 0.1, as justified in Appendix I. Ablation studies on $M$ and the hyper-encoder architecture can be found in Appendix C. For comparison, we also perform continual pretraining of the base model using LoRA under identical training conditions, serving as a baseline (denoted as Cont. Pretraining in the tables). The continual pretraining process is highly efficient, requiring approximately 10 H100-GPU hours for a 4B-parameter model and up to 40 H100-GPU hours for a 14B-parameter model, using only 0.1 billion training tokens. Interestingly, the *reconstruction loss* converges to near zero during continued pretraining, indicating that the model can almost perfectly recover the original base token sequences from the hypertoken representations. This highlights the learned compression is highly information-preserving. Details of the training setup, compute infrastructure, and dataset curation are provided in Appendices I and J.

## 3.2 Qualitative Examples and Hypertoken Patterns

We present several examples (Figure 4 and Table 1) to provide intuition into how the zip2zip model generates text. We see that the model generates a mixture of hypertokens and base tokens in the output (Figure 4). The hypertoken ratio is as high as 40% in the Python code generation example, and 20% in the biomedical text generation example. Many of the hypertokens correspond to semantically meaningful units or domain-specific terms as shown in Table 1. For a more fine-grained visualization of hypertoken with zip2zip, we provide visualizations of token streams in Figure 10 in the appendix.

## 3.3 Token Efficiency

Given an input text $x$ and a tokenizer, we define the *token efficiency* $\eta := \frac{\text{Bytes}(x)}{\text{Tokens}(x)}$ as the average number of bytes represented by each token (also called *compression ratio*), where $\text{Bytes}(x)$ refers to the number of bytes in the UTF-8 encoding of $x$. This measures how compactly a tokenizer encodes input text—higher values of $\eta$ indicate more efficient tokenization. We evaluate token efficiency using the tokenizers of four LLMs—Llama-3 [Grattafiori et al., 2024], Qwen-2 [Yang et al., 2024], Phi-4 [Abdin et al., 2024], and Gemma-3 [Team, 2025]—each associated with a different base vocabulary size ranging from 128K to 256K. Token efficiency is measured across five representative domains, sampled from publicly available datasets: code [Lozhkov et al., 2024b], math [LI et al., 2024], chat [Ding et al., 2023], multilingual [Penedo et al., 2024], and web [Lozhkov et al., 2024a]. Table 2 shows that applying LZW zip2zip consistently improves token efficiency across all tokenizer and domains. Gains are particularly strong in structured domains like code and math—with gains of 48% and more over the base tokenizer. Interestingly, models with larger vocabulary sizes do not always achieve better token efficiency, suggesting that simply enlarging the vocabulary size is not sufficient to improve it.

Table 2: Token efficiency (bytes per token) across domains for different tokenizers with and without `zip2zip`.

| Tokenizer | Code | Math | Chat | Multilingual | Web |
|---|---|---|---|---|---|
| Llama-3-128K [Grattafiori et al., 2024] | 4.1 | 2.7 | 5.1 | 3.8 | 4.6 |
| +zip2zip | 6.3 (+54%) | 4.0 (+48%) | 6.4 (+25%) | 4.7 (+24%) | 5.4 (+17%) |
| Qwen-2-150K [Yang et al., 2024] | 4.0 | 2.3 | 5.1 | 3.7 | 4.4 |
| +zip2zip | 6.2 (+55%) | 3.7 (+61%) | 6.4 (+25%) | 4.6 (+24%) | 5.2 (+18%) |
| Phi-4-200K [Abdin et al., 2024] | 4.1 | 2.7 | 5.4 | 4.6 | 4.7 |
| +zip2zip | 6.3 (+54%) | 4.1 (+52%) | 6.7 (+24%) | 5.5 (+20%) | 5.4 (+15%) |
| Gemma-3-256K [Team, 2025] | 3.3 | 2.3 | 5.0 | 4.4 | 4.5 |
| +zip2zip | 5.6 (+70%) | 3.7 (+61%) | 6.4 (+28%) | 5.4 (+23%) | 5.4 (+20%) |

## 3.4 Perplexity

We evaluate the perplexity of `zip2zip` models on four corpora: Wikitext [Merity et al., 2016], The Pile [Gao et al., 2020], and two subsets of Paloma [Magnusson et al., 2023]: mC4, a multilingual subset of C4, and dC4 (aka C4-100D), a subset of C4 spanning 100 domains. Given a token sequence $x = x_1, \ldots, x_N$, and a model $q$, perplexity and byte-level perplexity [Radford et al., 2019, Magnusson et al., 2023] are defined as: $\mathrm{PPL} := \left( \prod_{i=1}^{N} q(x_i) \right)^{-1/N}$, $\mathrm{Byte\text{-}PPL} := \left( \prod_{i=1}^{N} q(x_i) \right)^{-1/B} = \mathrm{PPL}^{1/\eta}$, where $B$ is the number of UTF-8 bytes of the text, and $\eta$ denotes the token efficiency (i.e., bytes per token). Token-level perplexity depends on the tokenization scheme and is unsuitable for cross-tokenizer comparison. We instead report byte-level perplexity, a vocabulary-agnostic metric that normalizes for tokenization differences. Table 3 (right panel) shows that `zip2zip` models see a modest increase in byte-level perplexity, indicating a slight drop in language modeling performance.

Table 3: Two-shot accuracy across seven NLP benchmarks (left) and byte-level perplexity ($\downarrow$) on four corpora using a 1024-token context window (right). Standard deviations (bootstrapped) $\approx 0.02$ across all tasks.

| Model | Method | ARC-c | ARC-e | HS | OBQA | PIQA | WG | GSM8K | Wiki | Pile | mC4 | dC4 |
|---|---|---|---|---|---|---|---|---|---|---|---|---|
| Phi-3.5-4B | Base | 0.60 | 0.83 | 0.66 | 0.46 | 0.79 | 0.75 | 0.82 | 1.58 | 1.79 | 1.88 | 1.74 |
| | Cont. pretrain | 0.60 | 0.82 | 0.63 | 0.47 | 0.82 | 0.75 | 0.40 | 1.59 | 1.81 | 1.88 | 1.74 |
| | zip2zip | 0.57 | 0.83 | 0.61 | 0.46 | 0.82 | 0.75 | 0.15 | 1.69 | 1.95 | 2.00 | 1.82 |
| Phi-3-14B | Base | 0.62 | 0.80 | 0.70 | 0.51 | 0.83 | 0.76 | 0.84 | 1.43 | 1.72 | 1.82 | 1.67 |
| | Cont. pretrain | 0.62 | 0.88 | 0.66 | 0.52 | 0.87 | 0.80 | 0.52 | 1.47 | 1.79 | 1.86 | 1.68 |
| | zip2zip | 0.62 | 0.86 | 0.68 | 0.51 | 0.85 | 0.79 | 0.25 | 1.56 | 1.90 | 1.96 | 1.75 |

## 3.5 Evaluation on NLP Benchmarks

We next evaluate `zip2zip`'s few-shot performance on real-world tasks. We evaluate on seven widely used NLP benchmarks, including ARC-[Challenge, Easy] [Clark et al., 2018], HellaSwag [Zellers et al., 2019], LAMBADA [Paperno et al., 2016], OpenbookQA [Mihaylov et al., 2018], PIQA [Bisk et al., 2019], Winogrande [Sakaguchi et al., 2019] and GSM8K [Cobbe et al., 2021]. As shown in Table 3, the model continued-pretrained with `zip2zip` performs similarly to the baseline on most tasks. However, on GSM8K, where the primary task involves numerical computation, the model exhibits significant degradation. While token-level operations are already known to be challenging for LLMs [Singh and Strouse, 2024], it is possible that adaptive tokenization exacerbates this effect, though further validation is required to confirm this hypothesis.

**Multilinguality.** To validate the effectiveness of `zip2zip` on non-English languages, we evaluate the model on machine translation tasks, including WMT14 [Macháček and Bojar, 2014], WMT16 [Bojar et al., 2016]. The results, shown in Table 4, indicate a small performance degradation across the BLEU, CHRF, and TER metrics when using `zip2zip`. However, the drop is relatively minor, suggesting that the model retains strong multilingual capabilities even in the compressed representation. Additional experiments on multilingual QA benchmarks are provided in Appendix H.2.

Table 4: Machine translation performance on WMT benchmarks. Scores are averaged across both translation directions. Standard deviations (approximately $1.0 \sim 2.0$) are reported in Table 11 in Appendix H.

| Model | Method | WMT14 En-Fr | | | WMT16 En-De | | | WMT16 En-Ro | | |
|---|---|---|---|---|---|---|---|---|---|---|
| | | BLEU↑ | CHRF↑ | TER↓ | BLEU↑ | CHRF↑ | TER↓ | BLEU↑ | CHRF↑ | TER↓ |
| Phi-3.5-4B | Base | 33.6 | 58.3 | 53.0 | 39.2 | 63.2 | 47.9 | 17.7 | 45.5 | 73.4 |
| | Cont. pretrain | 36.5 | 61.0 | 51.5 | 42.3 | 65.4 | 44.9 | 16.7 | 45.8 | 79.7 |
| | zip2zip | 34.1 | 59.4 | 54.5 | 39.7 | 64.5 | 48.0 | 14.3 | 44.2 | 93.5 |
| Phi-3-14B | Base | 39.1 | 62.6 | 49.3 | 43.1 | 65.6 | 44.1 | 21.3 | 51.0 | 70.5 |
| | Cont. pretrain | 38.9 | 63.2 | 48.8 | 48.4 | 70.1 | 39.8 | 21.8 | 52.0 | 68.3 |
| | zip2zip | 36.4 | 62.8 | 51.2 | 44.8 | 68.1 | 42.9 | 19.5 | 50.1 | 72.9 |

## 3.6 Inference Efficiency

`zip2zip` reduces decoding time by lowering the number of tokens that need to be generated. However, it introduces additional FLOPs due to the on-the-fly computation of hyper-embeddings by the hyper-encoder. To address this overhead, we implement hyper-embedding caching and optimize the computation using a custom Triton kernel. We report separate timings for *prefilling* and *decoding* across multiple models, with and without `zip2zip`, in Table 5. As we show in Table 5, `zip2zip` achieves a significant speedup in all four settings. Both prefilling and decoding times are significantly reduced, with the most substantial gains observed in the 512+256 setting with the Phi-3.5-4B model. Improvements are significantly stronger on datacenter-grade GPUs like the NVIDIA H100 and more modest on consumer hardware (e.g., Apple M1).

Table 5: **Throughput (tokens/sec)** comparison of the `zip2zip` framework against the baseline HuggingFace Transformers `generate` and MLX `generate` implementation. Performance is detailed for prefilling and decode phases across various context lengths (first value in column headers) combined with a 256-token generation length. `zip2zip` demonstrates notable throughput improvements, in both the prefilling and decoding phases.

| Setting | Method | 256+256 | | 512+256 | | 1024+256 | | 2048+256 | |
|---|---|---|---|---|---|---|---|---|---|
| | | Prefill | Decode | Prefill | Decode | Prefill | Decode | Prefill | Decode |
| *Hardware: Apple M1 (16GB RAM)* | | | | | | | | | |
| **Phi-3.5-4B** | Base model | 165.0 | 7.3 | 211.3 | 7.5 | 200.9 | 7.1 | 196.6 | 6.8 |
| | zip2zip | 145.5 | 7.9 | 231.4 | 10.1 | 189.6 | 7.4 | 233.8 | 7.3 |
| | **Relative %** | -11.8% | +7.5% | +9.5% | +34.8% | -6.6% | +3.9% | +18.9% | +7.5% |
| *Hardware: NVIDIA H100 80GB GPU* | | | | | | | | | |
| **Phi-3.5-4B** | Base model | 700.9 | 56.2 | 1347.2 | 54.4 | 2689.4 | 52.8 | 4993.2 | 53.1 |
| | zip2zip | 936.6 | 61.4 | 2722.1 | 79.8 | 4326.7 | 61.5 | 9258.1 | 61.9 |
| | **Relative %** | +33.6% | +9.3% | +102.6% | +46.6% | +60.9% | +16.6% | +85.4% | +16.5% |
| **Phi-3-14B** | Base model | 724.4 | 44.6 | 1356.3 | 43.8 | 2328.6 | 45.1 | 3849.5 | 42.2 |
| | zip2zip | 1024.6 | 54.9 | 1973.0 | 61.1 | 3657.0 | 66.8 | 7239.1 | 46.3 |
| | **Relative %** | +41.5% | +23.0% | +45.5% | +39.5% | +57.0% | +48.1% | +88.1% | +9.6% |

**Efficient LZW-Tokenizer Implementation.** `zip2zip` introduces an additional LZW compression step during inference and a decompression step at the end of generation. As a result, the efficiency of LZW-integrated tokenization is important to overall performance. To minimize overhead, we implemented a Rust-based `zip2zip` tokenizer that outperforms the Python version (see Figure 7) and matches the latency of HuggingFace's fast BPE tokenizer.

## 4 Related Work

**Domain-Adapted Tokenizers.** Several works have explored tokenizer adaptation by expanding the token vocabulary to better support specific domains or languages. Zhao et al. [2024], Kim et al. [2024], Liu et al. [2023, 2024a] adapt the Llama tokenizer to Chinese, Korean, and specialized domains such as mental health and law by adding new tokens. However, these approaches yield a fixed vocabulary that does not adapt during inference.

**Input Compression for LLMs.** Prompt compression methods such as gist tokens [Mu et al., 2023], selective context [Li et al., 2023], LLMLingua [Jiang et al., 2023], summary vectors [Chevalier et al., 2023], in-context autoencoders [Ge et al., 2024], and others [Wingate et al., 2022] reduce the context length by lossy compression. While their lossy compression nature enables high compression ratios, these prompt compression methods can only compress input tokens, but not the output tokens, although output tokens typically dominate generation time under low-batch workloads. Lester et al. [2024] propose improving language model efficiency by training LLMs directly on text compressed with arithmetic coding.

**Transformers with Dynamic Embeddings.** Architecture-wise, `zip2zip` employs a dynamic embedding layer built upon transformer blocks. Similar ideas have been explored in prior work aimed at reducing the computational cost of transformers, including Hourglass [Nawrot et al., 2022], dynamic-pooling transformer [Nawrot et al., 2023], MegaByte [Yu et al., 2023], Toucan [Fleshman and Durme, 2023], Learn-Your-Token [Thawani et al., 2023], SpaceByte [Slagle, 2024], ZeTT [Minixhofer et al., 2024], dynamic tokenization [Feher et al., 2025], BLT [Pagnoni et al., 2025], and H-Net [Hwang et al., 2025]. These approaches vary in their model architectures and chunking strategies. Dynamic Vocab [Liu et al., 2024b] is probably the closest in terms of conceptual motivation, as it also expands the vocabulary dynamically during generation. The main difference lies in the dynamic vocabulary construction algorithm and the model training procedure.

# 5 Discussion and Limitations

**Beyond LZW.** While we adopt LZW for dynamic construction of hypertokens, `zip2zip` is broadly compatible with any online compression algorithm. Future work may explore alternative schemes that provide different trade-offs between compression efficiency and model performance.

**Codebook Management Strategy.** The LZW algorithm grows the codebook linearly with the number of tokens in the context window. Empirical results show that only about 25% of hypertokens are reused during generation, leaving substantial room for optimization. Two potential improvements are (1) *pruning* or *selective retention* strategies to reduce unused entries, and (2) *codebook prefilling*, which could be beneficial if likely tokens can be speculated ahead of input processing.

**Optimization Under Lossless Compression.** Since `zip2zip` employs lossless compression, the achievable performance is theoretically invariant under the transformation: a *transported model*, as described in Section 2.6, can attain identical likelihood to that in the original space. Empirically, however, we observe a mild increase in perplexity under compression (Table 3), indicating that the trained model does not perfectly recover the *transported model*. This discrepancy arises from optimization challenges rather than representational limits—gradient descent may converge more slowly or settle in suboptimal regions due to more complex loss landscape. Understanding this optimization difficulty—how the search landscape changes under compression and whether targeted preconditioning or extended training budgets can close the gap—remains an important question for future work.

# 6 Conclusion

We presented `zip2zip`, an approach that brings inference-time tokenization to large language models through online token compression. By combining LZW-based sequence compression with dynamic hypertoken embeddings, `zip2zip` enables compact, adaptive tokenization with lightweight uptraining and little architectural changes. Across multiple domains and languages, it achieves substantial reductions in sequence length and decoding cost while maintaining strong task performance. These results demonstrate that online token compression can serve as a practical path toward dynamic tokenization, pointing to new directions for efficient and adaptable LLM inference.

# Acknowledgements

We would like to thank Miao Xiong for proof-reading the paper and providing valuable feedback. We are also grateful to Emre Kıcıman, Jason Eisner, Barun Patra, Ana-Maria Indreias, Xiuying Wei, Julian Minder, Tiago Pimental, Luca Beurer-Kellner, and Aleksei Kudrinskii for their helpful

input and insightful discussions throughout this project. Special thanks to Zheng Zhou for providing technical support with the computing infrastructure.

West's lab is partly supported by grants from Swiss National Science Foundation (TMSGI2_211379 and Grant 200364), Swiss Data Science Center (P22_08), H2020 (952215), Microsoft Swiss Joint Research Center, and Google, and by generous gifts from Facebook, Google, and Microsoft.

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

# A More Related Work

Wang et al. [2025], Liu et al. [2024a] conducted studies on how to effectively expand the vocabulary by better selecting the subset of tokens to add. Multi-Word Tokenizer [Gee et al., 2023] and SuperBPE [Liu et al., 2025] demonstrated that allowing forming tokens beyond word boundaries in BPE vocab learning helps to achieve more compact tokenization and even improves model performance. Content-Adaptive Tokenizer (CAT) by [Shen et al., 2025] introduces a dynamic image tokenization approach that allocates tokens based on content complexity, achieving improved reconstruction quality and efficiency compared to fixed-size tokenization methods. [Lotz et al., 2025] propose a pixel-level fallback encoder that bypasses subword vocabulary limitations by rendering text as images, enabling vocabulary-free representations that improve multilingual performance and efficiency in pretrained language models. The FlexiTokens [Owodunni et al., 2025] introduces learnable, byte-level tokenizers that dynamically adapt token boundaries to new domains and languages, reducing over-fragmentation.

# B   Discussions on Merge Size

## B.1   An Upper Bound on Merge Size

**Proposition B.1.** *Let $T$ be an input sequence of length $N$ over a finite alphabet. LZW compression algorithm merges substrings by identifying and replacing the most frequent substrings with new symbols, iteratively. Then, the size $M$ of the largest merged unit (i.e., the longest substring created via merging) is bounded above by $O(\sqrt{N})$.*

*Proof.* Assume that the largest merged unit has size $M$. This implies that there exists at least one merge at level $M$ involving a substring of length $M$. Furthermore, due to the nature of merge-based algorithms, any merged unit of size $k$ must be composed of previously merged units of smaller sizes (e.g., from sizes $k-1$ and 1, or similar). Hence, in order to construct a merged unit of size $M$, the algorithm must have previously created all merged units of sizes 1 through $M-1$.

Thus, the existence of a merged unit of size $M$ implies the existence of merged units of every size $k$ such that $1 \le k \le M$. Each such unit must occur at least once in the input sequence in order to be merged.

Therefore, the total number of characters in $T$ must be at least the sum of the lengths of all merged units from size 1 to $M$, i.e.,

$$N \ge \sum_{k=1}^{M} k = \frac{M(M+1)}{2}.$$

This implies:

$$M = O(\sqrt{N}).$$

Thus, the length $M$ of the largest merged unit is bounded above by $O(\sqrt{N})$.  □

## B.2   Relation Between Merge Size and Compression Rate

**Definition B.1** (Compression Rate). *We define the* compression rate *as the ratio between the number of tokens after compression ($N_{comp}$) and the number of tokens in the original uncompressed text ($N_{orig}$), expressed as a percentage:*

$$Compression\ Rate = \frac{N_{comp}}{N_{orig}} \times 100\%.$$

*A lower compression rate indicates greater reduction in token count, and thus more effective compression.*

The last column of Table 6 shows how the maximum merge size $M$ affects compression rate when the context window length is 2048. As $M$ increases, compression rate improves significantly, especially from $M = 1$ to $M = 3$. Beyond that, gains diminish, suggesting $M = 3$ strikes a good balance between efficiency and compression rate.

Table 6: **Effect of maximum merge size ($M$) on byte-level perplexity and compression rate.** Perplexity is measured for Phi-3.5-4B across four corpora with a 1024-token context window. Compression rate is evaluated over the training corpus with a 2048-token context. $M = 1$ corresponds to no compression.

| $M$ | Wiki | Pile | mC4 | dC4 | Compression Rate(%) |
|---|---|---|---|---|---|
| 1 | 1.62 | 1.70 | 2.00 | 1.91 | 100.00 |
| 2 | 1.96 | 2.21 | 2.55 | 2.22 | 75.30 |
| 3 | 1.72 | 1.84 | 2.15 | 2.00 | 71.21 |
| 4 | 1.71 | 1.84 | 2.14 | 1.99 | 68.93 |
| 5 | 1.72 | 1.84 | 2.14 | 1.99 | 68.41 |

Interestingly, the relationship between maximum merge size and training loss in Figure 5 as well as perplexity in Table 6 is non-monotonic. The baseline case with $M = 1$ (i.e., no zip2zip compression) yields the lowest perplexity overall, which is expected and consistent with prior findings that

Table 7: **Ablation of hyper-encoder architecture** on byte-perplexity ($\downarrow$) across four corpora using a 1024-token context window. Performance improves with increasingly expressive architectures.

| Model | Method | Wiki | Pile | mC4 | dC4 |
|---|---|---|---|---|---|
| Phi-3.5-4B | averaging | 1.81 | 1.97 | 2.29 | 2.08 |
| | 1-attention-layer | 1.73 | 1.86 | 2.16 | 2.01 |
| | 1-transformer-layer | 1.71 | 1.83 | 2.13 | 1.99 |
| | 2-transformer-layer | 1.72 | 1.84 | 2.15 | 2.00 |

compression typically incurs a trade-off in model performance. Among the compressed settings, the case $M = 2$ performs the worst, with noticeably slower convergence and higher final loss. In contrast, the case $M = 3$ achieves the best performance within the compressed configurations, striking a favorable balance between compression and prediction performance. While $M = 4$ and $M = 5$ also perform reasonably well, they exhibit slightly higher loss than $M = 3$, suggesting diminishing returns or possible over-compression at larger maximum merge sizes (see Figure 5).

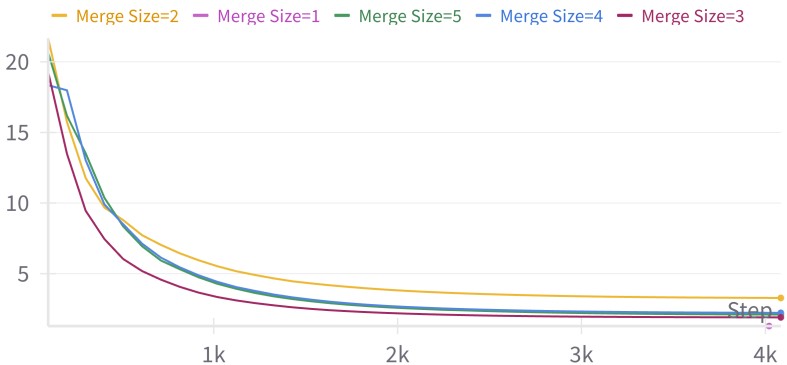

Figure 5: **Effect of maximum merge size $M$ on zip2zip training loss**: $M = 1$ (no compression) achieves the lowest loss overall. Among compressed settings, $M = 3$ performs best, while $M = 2$ shows the worst convergence. Larger $M$ (4 and 5) yield slightly worse results than $M = 3$.

Table 6 reports the byte-level perplexity across four corpora using a 1024-token context window. The results align closely with the training loss trends observed earlier. Setting $M = 1$ (i.e., no compression) consistently achieves the lowest perplexity across all datasets, reaffirming that compression introduces a performance trade-off. Notably, $M = 2$ performs the worst across all corpora, exhibiting the highest perplexity values. For merge sizes $M = 3$, $M = 4$, and $M = 5$, perplexity scores are nearly identical, suggesting that moderate compression can be achieved without significantly sacrificing language modeling quality—provided $M = 2$ is avoided. This consistency across loss and perplexity metrics further supports the robustness of maximum merge size $M = 3$ as the most effective trade-off point.

## C    Discussions on Hyper-Encoder Architecture

### C.1    Hyper-encoder architecture

We ablate the architecture of the hyper-encoder to evaluate its effect on language modeling performance, as shown in Table 7. We compare increasingly expressive architectures, starting with a simple averaging method that introduces no additional parameters. This baseline yields the highest perplexity, highlighting its limited capacity. Adding a single attention layer significantly improves performance, and further gains are observed with a 1-layer transformer encoder. The 2-layer transformer offers marginal additional benefit, suggesting that a lightweight transformer (1–2 layers) is sufficient for effective hypertoken modeling.

Figure 6 illustrates the effect of hyper-encoder architecture on zip2zip training loss. We observe that the simple averaging method converges the fastest but plateaus at a relatively high loss, reflecting its limited capacity. As model complexity increases—with attention and transformer layers—the convergence becomes slower, yet the final loss is significantly lower. Notably, the 1-layer and 2-layer

transformer encoders yield the best performance, demonstrating that additional parameters enable the model to better capture structure, albeit at the cost of slower training dynamics.

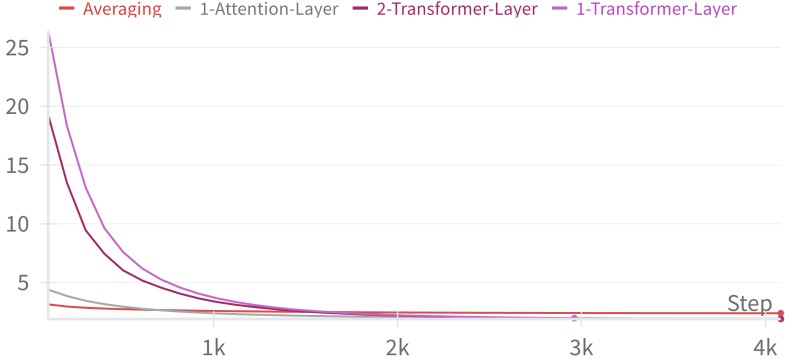

Figure 6: **Effect of hyper-encoder architecture on zip2zip training loss.** Averaging (no additional parameters) converges quickly but to a higher loss. As architectural complexity increases—from attention to transformer layers—convergence becomes slower, but the final loss is lower. This highlights a trade-off between training speed and modeling capacity.

## D   Discussions on Compression

Table 8: Token statistics for Code Generation, Biomedical, and French QA domains.

| Stats | Code | Biomedical | French |
|---|---|---|---|
| Original Seq Len | 344 | 322 | 399 |
| Zip2Zip Seq Len | 265 | 270 | 367 |
| Num Hypertoken | 44 | 35 | 26 |
| Hypertoken Ratio | 0.166 | 0.130 | 0.071 |
| Compression Rate | 0.770 | 0.839 | 0.920 |

Table 8 shows detailed token statistics on illustrative examples across three domains in Figure 4, highlighting zip2zip's ability to reduce sequence length and introduce reusable hypertokens with domain-specific efficiency.

Table 9: **Sequence length reduction** (%) across domains, inferred from the inverse of token efficiency gains in Table 2.

| Tokenizer | Code | Math | Chat | Multilingual | Web |
|---|---|---|---|---|---|
| Llama-3-128K | 34.9% | 32.5% | 20.3% | 19.1% | 14.8% |
| Qwen-2-150K | 35.5% | 37.8% | 20.3% | 19.6% | 15.4% |
| Phi-4-200K | 34.9% | 34.1% | 19.4% | 16.4% | 13.0% |
| Gemma-3-256K | 41.1% | 37.8% | 21.9% | 18.5% | 16.7% |

Table 9 reports the estimated sequence length reduction across domains, showing that zip2zip consistently shortens token sequences by 13–41%, with the strongest gains observed in structured domains like code and math.

## E   FLOPs Estimation for `zip2zip`

Following the assumptions of Kaplan et al. [2020], we estimate training FLOPs ($\Gamma$) as:

$$\Gamma \approx 6 \cdot N_{\text{tokens}} \cdot N_{\text{params}},$$

where $N_{\text{tokens}}$ is the total number of processed tokens and $N_{\text{params}}$ is the number of trainable parameters. This estimate ignores the quadratic attention cost, assuming:

$$12 \cdot d_{\text{model}} \ll \text{sequence length.}$$

For `zip2zip`, this becomes:

$$\Gamma_{\text{z2z}} \approx 6 \cdot N_{\text{tokens}} \cdot \rho \cdot N_{\text{params}}(1 + \alpha),$$

where $\rho$ is the compression ratio, and $\alpha$ accounts for the overhead of the hyper-encoder applied at the embedding and LM head. The relative FLOPs ratio is then:

$$\frac{\Gamma_{\text{z2z}}}{\Gamma} = \rho \cdot (1 + \alpha).$$

Assuming the hyper-encoder mirrors the base model's configuration, we estimate:

$$\alpha \approx \frac{lM}{L},$$

where $l$ is the number of hyper-encoder layers, $M$ is the maximum merge size, and $L$ is the number of base model layers. We illustrate this estimate across several model scales in Table 10, showing that the relative FLOPs overhead from the hyper-module remains modest (typically under 15%).

| Model | L | M | l | $\alpha = \frac{lM}{L}$ |
|---|---|---|---|---|
| Transformer-4B | 14 | 2 | 1 | 0.14 |
| Transformer-7B | 32 | 2 | 2 | 0.13 |
| Transformer-70B | 80 | 3 | 3 | 0.11 |
| Transformer-400B | 128 | 3 | 4 | 0.09 |

Table 10: Relative FLOPs overhead from the hyper-module across different model sizes.

## F  Discussions on Tokenizer

Figure 7 compares the tokenization and detokenization latencies across different tokenizer configurations. The Base Tokenizer corresponds to the standard BPE tokenizer implemented by the Hugging Face tokenizers library. The Rust LZW Tokenizer represents the end-to-end latency when LZW compression and decompression are applied on top of the BPE tokenization. As shown, this configuration introduces only a small additional latency in the tokenization process while leaving the detokenization latency virtually unchanged. The Python LZW Tokenizer, in contrast, exhibits significantly higher latency due to Python's runtime overhead. Overall, the results indicate that most of the observed latency arises from the BPE segmentation process itself rather than the LZW compression, suggesting that efficient implementations of compression add minimal overhead to tokenization workflows.

## G  Entropy Invariance under Lossless Transformations

**Theorem G.1** (Entropy Invariance under Lossless Compression)**.** *Let $g : \mathcal{X}^* \to \mathcal{Z}^*$ be a bijection onto its image, and let $P_Z$ and $p_\gamma$ denote the push-forward distributions of $P_X$ and $p_\theta$, respectively:*

$$P_Z(z) = P_X(g^{-1}(z)), \qquad p_\gamma(z) = p_\theta(g^{-1}(z)).$$

*Then the total entropy and cross-entropy are invariant under g:*

$$H(P_Z) = H(P_X), \qquad H(P_Z, p_\gamma) = H(P_X, p_\theta).$$

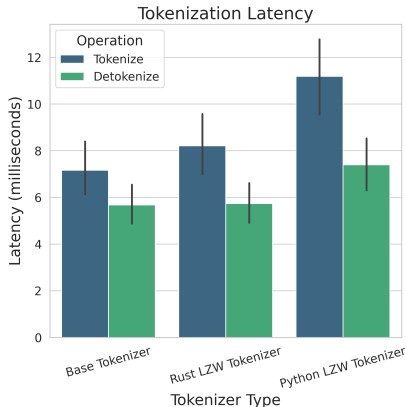

Figure 7: `zip2zip` tokenizer latency (ms) vs. HF tokenizer.

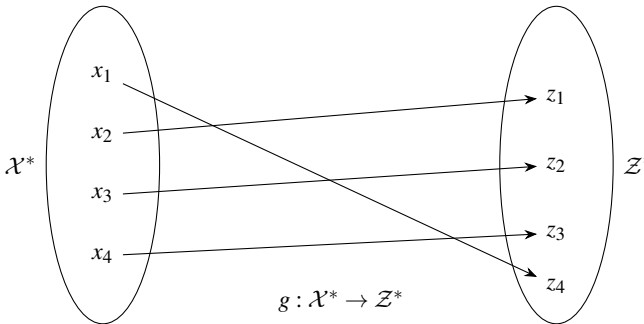

Figure 8: Lossless compression mapping (bijective) $g$ (e.g., LZW) from original sequences $\mathcal{X}^*$ to compressed sequences $\mathcal{Z}^*$.

*Proof.* By definition of the push-forward distribution,

$$
\begin{aligned}
H(P_Z) &= -\sum_{z \in \mathcal{Z}^*} P_Z(z) \log P_Z(z) \\
&= -\sum_{z \in \mathcal{Z}^*} P_X(g^{-1}(z)) \log P_X(g^{-1}(z)) \\
&= -\sum_{g(x') \in \mathcal{Z}^*} P_X(x') \log P_X(x').
\end{aligned}
$$

Since $g$ is a bijection, we can replace the summation by $x' \in \mathcal{X}^*$, which yields

$$
H(P_Z) = -\sum_{x \in \mathcal{X}^*} P_X(x) \log P_X(x) = H(P_X).
$$

An identical argument applies to the cross-entropy by considering $P_X(x) \log p_\theta(x)$ as a single function, since the invariance property does not depend on the specific form of the function being summed:

$$
\begin{aligned}
H(P_Z, p_\gamma) &= -\sum_z P_Z(z) \log p_\gamma(z) \\
&= -\sum_x P_X(x) \log p_\theta(x) = H(P_X, p_\theta).
\end{aligned}
$$

Hence, both entropy and cross-entropy remain invariant under any lossless (bijective) transformation $g$. $\qquad\square$

# H Additional Results

## H.1 Machine Translation

We report standard deviations for machine translation results across WMT benchmarks in Table 11, computed using the lm-evaluation-harness codebase.

Table 11: Machine translation performance on WMT benchmarks (BLEU↑, CHRF↑, TER↓) with standard deviations (±) from bootstrapped estimates. Scores are averaged across both directions.

| Model | Method | WMT14 En-Fr | | | WMT16 En-De | | | WMT16 En-Ro | | |
|---|---|---|---|---|---|---|---|---|---|---|
| | | BLEU | CHRF | TER | BLEU | CHRF | TER | BLEU | CHRF | TER |
| Phi-3.5-4B | Base | 33.6±2.1 | 58.3±1.4 | 53.0±1.7 | 39.2±1.9 | 63.2±1.6 | 47.9±1.8 | 17.7±1.5 | 45.5±1.3 | 73.4±2.4 |
| | Cont. pretrain | 36.5±2.2 | 61.0±1.6 | 51.5±1.8 | 42.3±1.8 | 65.4±1.4 | 44.9±1.7 | 16.7±1.4 | 45.8±1.5 | 79.7±2.3 |
| | zip2zip | 34.1±1.9 | 59.4±1.5 | 54.5±2.0 | 39.7±1.7 | 64.5±1.6 | 48.0±1.9 | 14.3±1.6 | 44.2±1.4 | 93.5±2.5 |
| Phi-3-14B | Base | 39.1±2.0 | 62.6±1.4 | 49.3±1.9 | 43.1±2.0 | 65.6±1.5 | 44.1±1.7 | 21.3±1.5 | 51.0±1.4 | 70.5±2.2 |
| | Cont. pretrain | 38.9±2.2 | 63.2±1.4 | 48.8±1.9 | 48.4±2.0 | 70.1±1.3 | 39.8±1.9 | 21.8±1.4 | 52.0±1.3 | 68.3±2.9 |
| | zip2zip | 36.4±2.1 | 62.8±1.5 | 51.2±1.8 | 44.8±2.1 | 68.1±1.6 | 42.9±1.8 | 19.5±1.5 | 50.1±1.3 | 72.9±2.6 |

## H.2 Multilingual QA Tasks

We evaluate Phi-3.5–4B on multilingual downstream tasks beyond translation, including TruthfulQA-2, HellaSwag, and Winograd across five languages (French, Spanish, Russian, Chinese, and Arabic). As shown in Table 12, the base model demonstrates strong cross-lingual generalization. Continued pretraining in the vanilla setting slightly reduces performance, likely due to domain drift, while the zip2zip variant performs similarly. Overall, the results indicate that the proposed multilingual adaptation strategy maintains competitive performance across diverse evaluation benchmarks.

Table 12: Evaluation on Multilingual Tasks in addition to Translation. We report results on **TruthfulQA-2**, **HellaSwag**, and **Winograd** across 5 languages (FR, ES, RU, ZH, AR) using lm-evaluation-harness.

| Model | Method | TruthfulQA-2 | | | | | HellaSwag | | | | | Winograd | | | | |
|---|---|---|---|---|---|---|---|---|---|---|---|---|---|---|---|---|
| | | FR | ES | RU | ZH | AR | FR | ES | RU | ZH | AR | FR | ES | RU | ZH | AR |
| **Phi-3.5–4B** | Base | 0.47 | 0.51 | 0.46 | 0.41 | 0.42 | 0.61 | 0.66 | 0.49 | NA | 0.44 | 0.70 | NA | 0.74 | 0.74 | NA |
| | Cont. Pretrain. Vanilla | 0.42 | 0.47 | 0.46 | 0.46 | 0.43 | 0.54 | 0.57 | 0.37 | NA | 0.27 | 0.76 | NA | 0.70 | 0.61 | NA |
| | Cont. Pretrain. zip2zip | 0.41 | 0.46 | 0.47 | 0.43 | 0.40 | 0.55 | 0.55 | 0.36 | NA | 0.27 | 0.77 | NA | 0.73 | 0.64 | NA |

# I Technical Details

## I.1 Model and Training Configuration

- **Pretrained Model:** microsoft/Phi-3-medium-4k-instruct
- **Sequence Length:** 1024
- **Total Batch Size:** 32,768 tokens
- **Learning Rate Schedule:** Cosine decay
- **Learning Rate Range:** Max = 3e-4, Min = 1e-5
- **LoRA rank and alpha value:** Both are 32
- **Training Steps:** 10,000
- **Validation Interval:** Every 100 steps
- **Checkpoint Interval:** Every 500 steps
- **Pytorch Model Compilation:** Enabled

## I.2 LoRA Configuration

- **Rank:** 16
- **Alpha:** 16
- **Target Modules:** qkv_proj, o_proj, gate_proj, down_proj, up_proj

### I.3 Loss Weighting Coefficient

Since both the language modeling loss and the auxiliary reconstruction loss are formulated as cross-entropy objectives over tokens, they are naturally on a comparable scale. For a sequence of $N$ base tokens, the number of hypertokens used in the auto-reconstruction loss is upper-bounded by $N$, while each hypertoken corresponds to at most $M$ base tokens in the reconstruction target. The loss weighting coefficient $\lambda$ primarily serves to fine-tune the relative importance of the auxiliary objective rather than to correct for scale mismatch. We set $\lambda = 0.1$, which yielded stable training. We did not perform extensive exploration over the value of $\lambda$, which we leave as an interesting direction for future work.

### I.4 System and Libraries

- **Hardware:** $4 \times$ NVIDIA A100-SXM4-80GB GPUs, 64-core CPU (128 threads)
- **Key Libraries:**
  - PyTorch >= 2.5.0
  - Transformers >= 4.47.0
  - Datasets <= 3.1.0
  - Accelerate >= 0.26.0

### I.5 Compute Resources

We report the compute resources used for training our models in Table 13. All training was conducted on internal servers equipped with NVIDIA H100 GPUs. We estimate GPU-hours by multiplying wall-clock training time by the number of GPUs used. No additional compute was used beyond the reported experiments; we did not perform parameter grid search, large-scale hyperparameter tuning, or exploratory runs that were excluded from the paper.

Table 13: Training compute resources for zip2zip experiments.

| Model | GPUs | Time | GPU Type | GPU-Hours |
|---|---|---|---|---|
| Phi-3-Medium (14B) | 4 | 15h 46m | NVIDIA H100 80GB | 63.0 |
| Phi-3.5-Mini (4B) | 2 | 7h 0m | NVIDIA H100 80GB | 14.0 |

### I.6 Evaluation

All evaluations complete within 1 hour on a single A100 GPU.

## J Data Mixture

To support effective fine-tuning, we construct a curated dataset with balanced representation across diverse domains, including code, mathematics, dialogue, general web content, and multilingual text. The final dataset contains approximately 1 billion compressed tokens.

Table 14 summarizes the constituent datasets and their respective proportions. A visualization of the dataset composition and sequence length characteristics is shown in Figure 9.

| Dataset | Domain | Portion |
|---|---|---|
| HuggingFaceFW/fineweb-edu[Lozhkov et al., 2024a] | Web | 20% |
| devngho/the-stack-llm-annotations-v2[Lozhkov et al., 2024b] | Code | 25% |
| AI-MO/NuminaMath-1.5[LI et al., 2024] | Math | 20% |
| HuggingFaceH4/ultrachat_200k[Ding et al., 2023] | Chat | 20% |
| HuggingFaceFW/fineweb-2[Penedo et al., 2024] | Multilingual | 15% |

Table 14: Training data composition across domains.

The multilingual subset in `fineweb-2` includes the following languages: Mandarin Chinese (`cmn_-Hani`), German (`deu_Latn`), Japanese (`jpn_Jpan`), Spanish (`spa_Latn`), French (`fra_Latn`), Italian (`ita_Latn`), Portuguese (`por_Latn`), Dutch (`nld_Latn`), and Arabic (`arb_Arab`).

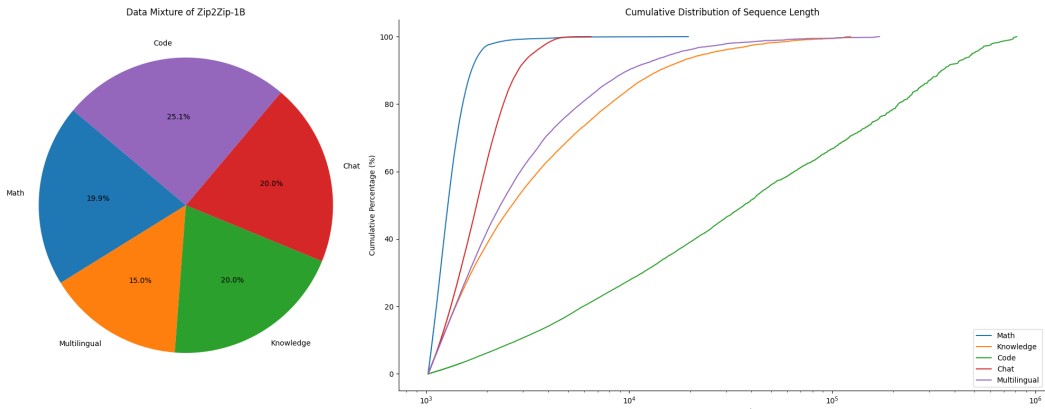

Figure 9: Left: Proportional breakdown of the fine-tuning dataset across five domains. Right: Cumulative distribution of input sequence lengths per domain (log scale). Code and multilingual data exhibit longer tail distributions, indicating greater variability in sequence lengths.

# K    Token Stream Visualization

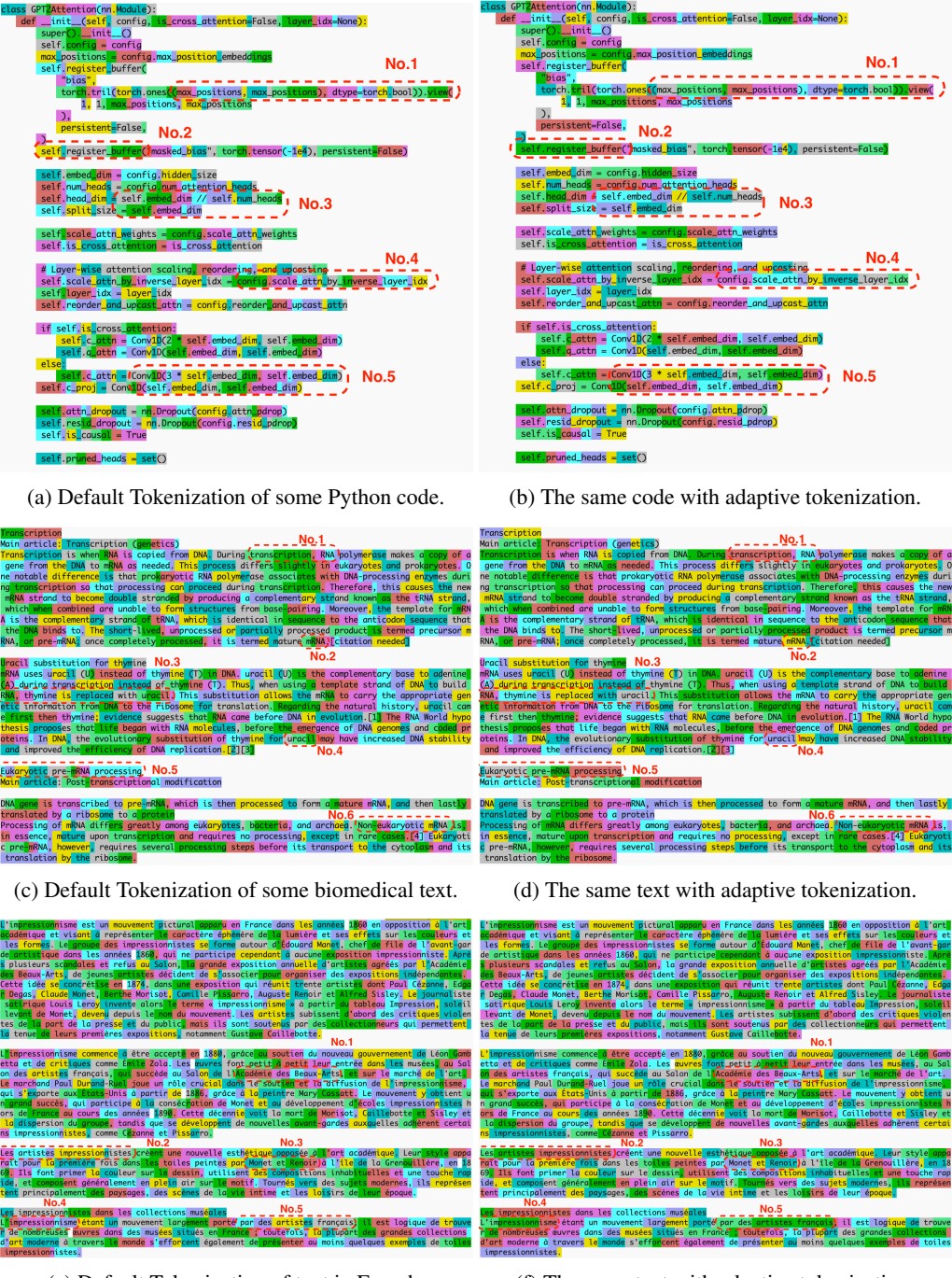

(a) Default Tokenization of some Python code.

(b) The same code with adaptive tokenization.

(c) Default Tokenization of some biomedical text.

(d) The same text with adaptive tokenization.

(e) Default Tokenization of text in French.

(f) The same text with adaptive tokenization.

Figure 10: Examples comparing default and adaptive tokenization. Dotted-line frames highlight where the differences are most noticeable.

