# OpenReview forum: "zip2zip: Inference-Time Adaptive Tokenization via Online Compression"
_NeurIPS.cc/2025/Conference — NeurIPS 2025 poster_

### Official Review · Reviewer_UZ7U · 2025-06-23

**Clarity:** 3
**Significance:** 3
**Originality:** 3
**Rating:** 4
**Confidence:** 4

**Summary:**

This paper proposes an adaptation in which the tokens are compressed into dynamic “hypertokens”. The paper evaluates performance changes on a variety of standard tasks as well as translation. The proposed method improves inference efficiency, but does negatively impact performance on some tasks.

**Questions:**

* Given the massive speedup after zipping, why not speed up training too (either through BLT-like changes or through superword tokenizers)?
* As math was the domain that showed the greatest drop in performance, could you modify the approach so that hyperwords involving numbers cannot be learned? This could prevent long sequences of numbers from being created that the model cannot work with well. I recommend looking into the number (pre-)tokenization literature [3-5].

[3] https://aclanthology.org/2024.emnlp-main.40/

[4] https://arxiv.org/abs/2402.14903

[5] https://arxiv.org/abs/2410.19730

Typos:
* L59: Phi-3-4B and Phi-3-14B but in the tables, the 4B model is Phi 3.5
* L225: needs a period after the close parenthesis
* L319-320: include → including

**Ethical Concerns:**

["NO or VERY MINOR ethics concerns only"]

**Final Justification:**

The authors have addressed several of my concerns. I think the points raised in the rebuttal, if added to the paper, will strengthen the arguments. However I still disagree about the use of 'inference-time' in the core framing of the paper. Therefore, I will keep my score as it was.

**Limitations:**

No. I think it would be helpful for the authors to address the possibility that this method worsens bias and toxicity (e.g. race and gender bias).

**Paper Formatting Concerns:**

None.

**Quality:**

3

**Strengths And Weaknesses:**

**Strengths**

This paper tackles a novel approach to a topic that has been the topic of recent popular work. In contrast to novel architectural approaches, this method proposes adaptation to existing models that speed up inference.

The paper is very clearly written. Examples provided in Figure 3 and Table 1 are extremely helpful for getting a sense of the impact of the method.

The methodology seems technically sound. The authors open source their code so it can be used and built on by others, making it a potentially valuable contribution.

**Weaknesses**

* Referring to this method as “inference-time” adaptation is somewhat misleading, as there is a necessary training stage with a non-negligible compute requirement.
* Comparing the zip2zip method only to base and continuous fine tuned models (Tables 2, 3, 4) may not be the most helpful comparison. I think a valuable comparison would be to compare to quantization. This would help contextualize the drop in performance and help the reader evaluate how changes in performance compare.
* Gains in token efficiency are the highest in code and math domains, but math is also the domain that shows the greatest drop in performance. This could suggest that the method is limited by a strong tradeoff between token efficiency and lower performance. This would have to be much better understood in order for other people to adopt the method. As a result, the claim in L358-360 that “zip2zip maintains strong performance across a range of tasks while achieving significant gains in inference efficiency” should be qualified, as improved inference efficiency seems to be correlated with a greater drop in performance.
* Table 2 shows the improved token efficiency in chat templates and multilingual data, but there are no evaluations in those domains. The authors should add more evaluations in these domains to help understand the relationship between increased token efficiency and changes in task performance. The inclusion of translation task results does not necessarily speak to multilingual performance, as translation is separate from within-language performance on non-English languages.
* The authors do not make any comparison to superword tokenizers [1-2]. In practice, the hypertokens look very much like superword tokens, so it would be helpful to draw a comparison and indicate the advantage of zip2zip is over static superword tokenization. For example, if you did some other type of standard vocabulary adaptation to turn a model’s tokenizer into a superword tokenizer, would you expect differences in terms of token efficiency or model performance relative to zip2zip?

[1] https://arxiv.org/abs/2503.13423

[2] https://arxiv.org/abs/2504.00178?

---

> ### Author Rebuttal · Authors · 2025-07-30
>
> We appreciate the reviewer’s thoughtful feedback and are encouraged by their recognition of our approach’s novelty, clarity, and potential utility through open-sourcing.Below we address your questions:
>
> ## W1. Referring to this method as “inference-time” adaptation is somewhat misleading as there is a necessary training stage
>
> We appreciate the reviewer’s concern regarding the phrasing of “inference-time” adaptation. To clarify: **zip2zip is explicitly designed for inference-time adaptation**. The one-time training step presented in the paper is **not due to a need for continual fine-tuning**, but rather reflects our desire to start from an existing pretrained LM (e.g., Phi or LLaMA), rather than training a model from scratch.
>
> Since these base models are not natively trained in the compressed zip2zip token space, we introduce a **single adaptation phase** to make them compatible with hyper-tokenized inputs. Crucially, this is **not task-specific or domain-specific fine-tuning**, and once performed, it enables downstream users to **deploy zip2zip without any additional training**, benefiting immediately from inference-time speedups.
>
> Looking ahead, our longer-term vision is for future language models to be **pretrained directly in the zip2zip space**, eliminating even this initial adaptation and enabling **fully training-free inference-time compression** from the outset.
>
> We hope this clarifies our intent behind the term and the design of zip2zip.
>
> ## W2. Comparing against quantization to provide a better understanding on efficiency VS performance
>
> We view **quantization and zip2zip as complementary techniques** that target different axes of model efficiency. While quantization reduces the **precision of model weights and activations**, zip2zip focuses on **input compression and reducing sequence length**, thereby decreasing the number of decoding steps required. These methods are inherently **orthogonal** and can be seamlessly combined.
>
> In practice, our implementation **trivially supports quantized models** (e.g., 4-bit and 8-bit), and we have observed that **combining zip2zip with quantization yields additive gains in inference efficiency**—without introducing conflicts or degradation in performance.
>
> ## W3: On the Tradeoff Between Token Efficiency and Model Performance
>
> We thank the reviewer for raising this important concern. We would like to share an **updated result** that directly addresses this point.
>
> Following a suggestion from Reviewer 1 regarding **segmentation ambiguity during evaluation**, we implemented a **multi-view perplexity evaluation** that accounts for multiple valid LZW segmentations of a given output string. This refinement addresses a key challenge in evaluating zip2zip: unlike standard tokenizations where a string maps to a single token sequence, zip2zip can yield **multiple compressed sequences that decode to the same text**, making canonical perplexity measurements potentially misleading.
>
> To resolve this, our new evaluation considers alternative valid segmentations (pruned in a beam-search-like fashion for tractability), leading to a **substantial improvement in perplexity**. As shown below, we recover **over 50% of the original gap** compared to the base model, demonstrating that much of the apparent performance drop was an artifact of rigid evaluation rather than a fundamental limitation:
>
> | **Model**     | **Method**         | **Wiki** | **Pile** | **mC4** | **dC4** |
> |---|---|-|-|-|-|
> | Phi-3.5-4B     | Base                | 1.62     | 1.88     | 1.94    | 1.77    |
> |               | Cont. finetune      | 1.63     | 1.89     | 1.94    | 1.77    |
> |               | zip2zip             | 1.71     | 2.02     | 2.04    | 1.84    |
> | **NEW**       | **zip2zip (multi-view)** | **1.66**     | **1.92**     | **1.97**    | **1.79**    |
>
> These results indicate that the performance-efficiency tradeoff is smaller than originally reported as multi-view evaluation provides a more faithful view of model capability. That said, we acknowledge that a residual gap remains—particularly on reasoning-heavy tasks—and we view this as a valuable direction for future work.
>
> P.S. An exmaple on how **multi-view perplexity** works
>
> #### base tokenization
> `[to, be, or, not, to, be, or, to, be, or, not]`
>
> #### canonical LZW tokenization
> `[to, be, or, not, tobe, or, tobeor, not]`
>
> #### non-canonical yet valid tokenization (multi-view)
> - `[to, be, or, not, to, be, or, tobeor, not]`
> - `[to, be, or, not, to, be, or, tobe, or, not]`
> - `[to, be, or, not, to, be, or, to,beor, not]`
> - `[to, be, or, not, to, be, or, to,be, or, not]`
>
> Multi-view perplexity evaluation sums the probabilities across all valid tokenizations that decode to the same output. This gives a more accurate picture of how likely the model generate the ground truth text. To keep the computation tractable and avoid exponential growth in the number of sequences, we branch over **each hypertoken individually**, rather than enumerating all possible joint segmentations as an approximation.
>
> ## W4. Evaluation on Multilingual Tasks In addition to Translation
>
> We have conducted additional experiments across **multilingual downstream tasks**, focusing on 5 languages (FR, ES, RU, ZH, AR). These include **TruthfulQA-2**, **HellaSwag**, and **Winograd**, all adapted for multilingual evaluation using [lm-evaluation-harness](https://github.com/EleutherAI/lm-evaluation-harness).
>
>
> | Model        | Method         | Truthful-QA-2 |        |||         | Hellaswag     |   ||| | Winograd          |      ||           | |
> |--|--|--|-|-|--|---|-|-|-|-|-|-|-|-|-|-|
> | | | FR  | ES     | RU            | ZH            | AR            | FR             | ES            | RU            | ZH            | AR            | FR             | ES            | RU            | ZH            | AR|
> | **Phi-3.5–4B** | Base     | 0.47     | 0.51         | 0.46         | 0.41         | 0.42         | 0.61     | 0.66    | 0.49     | NA            | 0.44      | 0.70    | NA     | 0.74    | 0.74    |NA    |
> |  | Cont. Pretrain. Vanilla | 0.42          | 0.47         | 0.46         | 0.46         | 0.43         | 0.54     | 0.57   | 0.37    | NA            | 0.27     | 0.76    | NA     | 0.70     | 0.61     |NA  |
> |           | Cont Pretrain. zip2zip | 0.41          | 0.46         | 0.47         | 0.43         | 0.40         | 0.55        | 0.55        | 0.36        | NA            | 0.27         | 0.77       | NA            | 0.73        | 0.64        |NA  |
>
>
> 1. Continued pretraining (vanilla) serves as a direct comparison point, as it was trained on the same corpus as the zip2zip variant.
>
> 2. The “N/A” entries indicate languages that are not currently supported for those tasks in [lm-evaluation-harness](https://github.com/EleutherAI/lm-evaluation-harness/tree/main).
>
> 3. The observed trends are consistent with the main task evaluation table reported in the paper
>
> ## W5. Comparison to Superword Tokenizers
>
> We see **zip2zip and superword tokenization as complementary techniques** rather than competing ones. Superword tokenizers aim to build a **stronger base vocabulary** at training time, improving efficiency by capturing frequent patterns statically. In contrast, zip2zip performs **dynamic, context-aware compression at inference time**, adapting on the fly to the specific patterns present uniquely in each input sequence.
>
> We agree that a comparison to **superword tokenizers** [1–2] would provide valuable insight for readers, especially given the apparent similarity between hypertokens and subword units with extended granularity. Due to time constraints, we were unable to include this comparison within the rebuttal window, but we see this as a **promising direction for follow-up work**. In fact, we are particularly interested in exploring whether zip2zip can be applied **on top of** superword tokenizers to further boost efficiency.
>
>
> ## Q1. Does zip2zip speed up training too?
>
>
> Yes—zip2zip can also accelerate training. When run on the same corpus, **zip2zip results in shorter effective context windows** due to improved segmentation and **larger token granularity**, allowing more text to be processed per training step. This leads to **faster throughput** and **reduced compute cost** during training.
>
> An additional practical advantage is that **zip2zip does not require training from scratch**. As demonstrated in our paper, we perform **continued pretraining** on top of an existing model (e.g., Phi or LLaMA) to adapt it to the compressed zip2zip space. In contrast, methods like **BLT** and **H-Net** are incompatible with existing tokenizers and therefore **require full training from scratch**, which significantly increases cost and limits accessibility.
>
> ## Q2. Can we disable merges of digit tokens to improve the performance on math ?
>
> Thank you for the thoughtful suggestion. We agree that finer-grained control over numeric token merges could be beneficial, especially given the observed performance drop in math-heavy domains.
>
> As you pointed out, forming long numeric hypertokens can make it harder for the model to reason effectively. In our current implementation, we already support **two control mechanisms** that help mitigate this issue:
> (1) a **maximum merge size** that limits how long a hypertoken can become, and
> (2) a **vocabulary cap** that restricts the number of hypertokens maintained during inference.
>
> The max merge size, in particular, can help prevent overly long digit sequences from being compressed into a single token. That said, we agree that **more targeted control—such as disabling merges over numeric spans entirely—would be a promising extension**. We appreciate the references to relevant work on number (pre-)tokenization [3–5], and we plan to explore this direction further in future iterations.
>
> ---
> We hope to have been able to address all the reviewers’ concerns, are happy to answer any follow-up questions you might have, and are looking forward to your reply.

---

> > ### Comment · Reviewer_UZ7U · 2025-08-04
> > **Response to Authors**
> >
> > >“inference-time” adaptation
> >
> > The authors describe the method as "a training pipeline for LZW compression-based finetuning" (L66-67) in the paper.  Later in this rebuttal, the authors use the term continued pretraining also to refer to zip2zip. In my mind, fine-tuning and continued pretraining do not constitute inference-time adaptations.
> >
> > >We view quantization and zip2zip as complementary techniques
> >
> > The fact that quantization and zip2zip combined lead to additive performance gains does not necessarily address my concern. Is it the case that quantization and zip2zip lead to comparable increases in efficiency? Do they have similar efficiency-performance tradeoffs? This comparison could help practitioners decide which method to use.
> >
> > >combining zip2zip with quantization yields additive gains in inference efficiency—without introducing conflicts or degradation in performance
> >
> > Are these results in the paper? I think this would strengthen the paper, especially in addition to a direct comparison between only quantization and only zip2zip.
> >
> > >Comparison to Superword Tokenizers
> >
> > I think it would strengthen the paper to include discussion on this point.
> >
> > >This leads to faster throughput and reduced compute cost during training.
> >
> > It would strengthen the paper to include this point and some figures on the extent of the speedup.

---

> ### Author Response · Authors · 2025-08-05
> **Clarifying the Inference-Time Adaptation Nature of Zip2Zip**
>
> We thank the reviewer for their thoughtful engagement and for participating in the rebuttal phase. We appreciate the
> opportunity to further clarify our terminology and intent.
>
> > The authors describe the method as "a training pipeline for LZW compression-based finetuning" (L66-67) in the paper. Later in this rebuttal, the authors use the term continued pretraining also to refer to zip2zip. In my mind, fine-tuning and continued pretraining do not constitute inference-time adaptations.
>
> We think this is a major misunderstanding that we feel obliged to explain in details.
>
> Terminology-wise, there are a few commonly used but not fully standardized terms:
>
> - finetuning VS continual-pretraining VS post-training
>
> as well as
>
> - inference-time adaption VS test-time adaption
>
> While these may carry slightly different meanings depending on the context, in our case, they refer to the same underlying concept. We use them interchangeably, aiming to clarify rather than confuse the reader.
>
> Let's breakdown into two questions:
>
> ## What is **Inference/test-time adaption** ?
>
> While there is no formal definition of inference- or test-time adaptation methods, many works have described them as…
>
> 1. *Test-time adaptation refers to the process of adapting models to testing data that may have distributional differences from the training data* [2]
> 2. *Test-time adaptation allows the model to adapt to the test data (i.e., target domain) in a source-free and online manner.* [7]
> 3. *Test-time adaptation (TTA) aims to adapt models towards test data to overcome the performance degradation caused by distribution shifts* [6]
> 4. *Several recent studies introduce a new paradigm called test-time adaptation for mitigating the domain shift. The idea of test-time adaptation is well aligned with real-world scenarios where a model needs to adapt to new environments quickly* [4]
>
> ---
>
> There is general agreement that **inference- or test-time adaptation** is the technique to *adapt the model to an input domain that is not known in advance*, in order to *mitigate domain shift*.
>
> In our case, the mismatch is between the *pretraining token distribution* and the *runtime token distribution*, which is often prompt-specific. As a result, the tokenizer vocabulary—typically optimized on a general corpus—does not align well with the domain encountered at test time. For instance, a general-purpose tokenizer tends to **fragment biomedical or low-resource language text**, reflecting the effects of domain shift.
>
> **Zip2Zip** is an *inference-time adaptation method* for the tokenizer. It dynamically adjusts the token vocabulary based on the input text, resulting in *input-specific segmentation* that better matches the target domain as we showed in Figure 3 and Table 1
>
> ---
>
> ## Does fine-tuning and continued pretraining constitutes inference-time adaptations?
>
> There are several approaches to achieving **inference-time adaptation**, with two notable categories:
> - **Test-time training**: In this setting, part of the model’s weights are updated during test time to adapt to the target domain [1][2][3][4].
> - **One-time post-training + test-time adaptation**: This line of work performs a *post-training* phase after pretraining, keeping the model weights frozen thereafter. The model is then capable of adapting to new test-time inputs without any further weight updates [5][6].
>
> **Zip2Zip** falls into the second category: it involves a one-time **post-training** step, after which the model can dynamically adapt to different inputs at test time—*without requiring any model updates*.
>
> Let's briefly examine how [5] and [6] implement **inference/test-time adaptation**:
>
> - **[5]** uses a combination of *SFT*, *DPO*, and *synthetic data* as post-training to improve instruction-following behavior. This enables the model to adapt to different safety configurations at inference time—without updating model weights.
>
> - **[6]** introduces a modified BatchNorm module with extra learnable interpolation weights. These extra weights are optimized in a *post-training* phase (after pretraining, before inference), allowing the model to adapt to new domains at test time while keeping the main model frozen, as stated in the paper:
> > We optimize the interpolating weight after the pre-training but before the test time, which we refer to as the post-training phase.
>
>
> ---
>
> ## References
>
> - [1] Parameter-free Online Test-time Adaptation (CVPR 2022)
> - [2] Efficient Test-Time Adaptation of Vision-Language Model (CVPR 2024)
> - [3] Frustratingly Easy Test-Time Adaptation of Vision-Language Models (NeurIPS 2024)
> - [4] SwapPrompt: Test-Time Prompt Adaptation for Vision-Language Models (NeurIPS 2023)
> - [5] Controllable Safety Alignment: Inference-Time Adaptation to Diverse Safety Requirements (ICLR 2025)
> - [6] TTN: A Domain-Shift Aware Batch Normalization in Test-Time Adaptation (ICLR 2023)
> - [7] EcoTTA: Memory-Efficient Continual Test-time Adaptation via Self-distilled Regularization (CVPR 2023)

---

> ### Author Response · Authors · 2025-08-05
> **Follow-up reply**
>
> We would like to express our gratitude to the reviewer again for the valuable feedback. We would like to calrify that we are not in a position to urge reviewer to raise the rating, we just want to clarify the points the reviewer raised above. Don't get us wrong.
>
> ---
>
> > I think it would strengthen the paper to include discussion on superword tokenizers on this point.
>
> We agree with the reviewer's comment that additional comparison to **superword tokenizers** [1][2] could provide additional insights for readers on how to choose between different techniques. However, we unfortunately will not be able to provide such results within the **rebuttal period**.
>
> That said,  **superword tokenizers are a very recent development**. The two papers mentioned by the reviewer [1][2] were made public **after March 1st, 2025**, and are therefore considered concurrent work relative to the NeurIPS submission deadline. As per NeurIPS guidelines, authors are **not expected to compare against concurrent work**.
>
> To be clear, we are not suggesting that these works are irrelevant—only that we are unable to provide empirical comparisons within the current timeframe. That said, they represent an interesting direction for future work. Given that superword tokenizers share a similar goal with Zip2Zip, i.e. reducing sequence length, we will try to have some comparison in the final manuscript or at least include them as related work to better guide readers.
>
> ---
>
> > It would strengthen the paper to include results on speedup of training and some figures on the extent of the speedup.
>
> We will not have empirical results on training speedup within the rebuttal phase, but we can provide a theoretical estimation. This theoretical speedup corresponds to the **compression rate of the data** used during training, which depends heavily on the dataset characteristics.
>
> As shown in the paper (Table 2), the **token efficiency**—a proxy for compression rate—indicates the expected speedup when using Zip2Zip during training.
>
> Let **r** be the **reduction rate**, defined as the ratio between the compressed sequence length and the original sequence length. The **theoretical training time** is proportional to both the time per step and the total number of training steps.
>
> Assuming the time per step remains constant (which it typically does), reducing the sequence length by a factor of **r** implies that we need fewer steps to see the same amount of content.
>
> For example, if `r = 0.6`, this would suggest a **40% reduction** in the total number of training steps—leading to a proportional reduction in overall training time.
>
>
> ## References
> - [1] https://arxiv.org/abs/2503.13423
> - [2] https://arxiv.org/abs/2504.00178

---

> > ### Comment · Reviewer_UZ7U · 2025-08-06
> > **Response**
> >
> > >We think this is a major misunderstanding that we feel obliged to explain in details.
> >
> > I think I still do not agree with the authors' labelling this method as inference-time adaptation, but do not believe we can achieve a better mutual understanding in this rebuttal.
> >
> > >As per NeurIPS guidelines, authors are not expected to compare against concurrent work.
> >
> > I understand. I think just including the points you mentioned in the previous reply would add to the paper. I don't think anything more involved than that should be expected.
> >
> > >We will not have empirical results on training speedup within the rebuttal phase, but we can provide a theoretical estimation.
> >
> > This is also understandable. If you happen to have those results, they would be nice to include in the final paper.
> >
> > While some of my concerns were addressed, I am still unsatisfied by the discussion of the inference-time terminology. I will keep my score, which still recommends accept.

---

### Official Review · Reviewer_tWZT · 2025-06-28

**Clarity:** 2
**Significance:** 2
**Originality:** 3
**Rating:** 5
**Confidence:** 3

**Summary:**

This work aims to improve the inference latency of LLMs by expanding the token vocabulary during the inference stage, thereby reducing the input and output sequence length.

**Questions:**

1 In Figure 5 and Table 7 of the Appendix, M=2 has the worst convergence and perplexity. This is an interesting phenomenon—do you have any possible explanations for this experimental result?

2  In Appendix D (Technical Details), the sequence length is set to 1024. Would increasing the sequence length affect the effectiveness of the results? Additionally, can the proposed methods help the model address issues caused by overly long sequences?

3 The paper lacks discussion on some works that also focus on adaptive tokenization, such as [1] and [2].


[1] Enhancing large language models through adaptive tokenizers NeurIPS (2024)

[2] Task-adaptive tokenization: Enhancing long-form text generation efficacy in mental health and beyond EMNLP (2023)

**Ethical Concerns:**

["NO or VERY MINOR ethics concerns only"]

**Final Justification:**

The authors have clearly addressed the concerns I raised. The paper is supported by extensive experiments and targets practical problems, offering valuable insights for future research in this direction. Therefore, I am raising my score to 5.

**Limitations:**

yes.

**Paper Formatting Concerns:**

no formatting concerns

**Quality:**

3

**Strengths And Weaknesses:**

Strengths:

1 The paper is well-structured and presents a high degree of originality by approaching faster inference from the perspective of tokenizer design.

2  It provides extensive experimental results to validate the proposed method, along with sufficiently detailed descriptions of the experimental setup.

3 The authors also release the code.

Weaknesses:

1 On certain tasks, such as GSM8K, there is a noticeable drop in performance.

2  This method introduces several additional components, such as the hyper-encoder and online vocabulary expansion, which may pose challenges for real-world deployment.

---

> ### Author Rebuttal · Authors · 2025-07-29
>
> Thank you for your thoughtful review and for highlighting the originality of our work and the extensive experimental validation.
> Below we address your questions:
>
> ## W1. On certain tasks, such as GSM8K, there is a noticeable drop in performance.
>
> This is a limitation we acknowledge in the paper, and we do consider it an important direction for future work to better understand and mitigate this gap.
>
> ## W2. Additional components, such as the hyper-encoder and online vocabulary expansion pose challenges for real-world deployment.
>
> Thank you for highlighting concerns regarding **deployment complexity** due to the added components in our method. While **zip2zip** introduces a **hyper-encoder** and **online vocabulary expansion**, we’ve carefully designed the system to ensure **minimal integration overhead** and **ease of adoption**:
>
> ###  Tokenizer Integration
> - Our compression mechanism is fully integrated with the tokenizer stack.
> - The API for `tokenizer.encode`, `tokenizer.decode`, `batch_encode`, and `batch_decode` remains **unchanged** and **compatible** with HuggingFace’s `tokenizers` library.
> - This allows plug-and-play use without altering preprocessing pipelines.
>
> ###  Modular Architecture
> - We provide a custom `DynamicEmbedding(nn.Module)` that encapsulates the complexity of embedding standard and hyper-tokens.
> - The **hyper-encoder** logic is cleanly isolated inside this module, which **replaces** only the **embedding** and **lm_head** layers—**all other model components remain untouched**.
>
> ### Codebook Management
> - We introduce a `CodebookManager` module that attaches to the model and acts as a **context manager** for tracking and managing **dynamic vocabulary expansion** during inference.
> - It supports **efficient updates** to the codebook and can operate across batch boundaries.
>
> ### High Efficiency
> - Embedding computations are backed by a **custom Triton kernel**, yielding performance **comparable to `torch.compile`**.
> - This ensures that zip2zip maintains **high inference throughput** without introducing latency bottlenecks.
>
> ---
>
> In summary, zip2zip is designed to make it a **drop-in enhancement** to existing model stacks, preserving **API compatibility**.
>
>
> ## Q1. Any explanation that smaller merge size leads to worse convergence ?
>
> Our hypothesis is that setting M as a hard maximum merge size introduces an abrupt boundary, which might be difficult for the model to learn. In contrast, with larger M, there are fewer interventions, and the compressed sequences behave more like natural LZW outputs. That said, this remains an unverified hypothesis.
>
> ## Q2.Can sequence length go easily beyond 1024 (experiment setting)?
>
> The sequence length of 1024 used in Appendix D was chosen based on practical hardware constraints—it reflects a realistic maximum context length for our available compute during training. We do not think there is anything in the zip2zip framework that would break with longer sequences. One consideration is the linear growth of the hyper-vocabulary  w.r.t. sequence length, which is not a real issue as we explained in our response to reviewer eQ7D in W5.
>
> Regarding issues caused by overly long sequences: Could you clarify which specific challenges you are referring to? (e.g., memory, latency, degradation in attention quality)
> One of the core advantages of zip2zip is its ability to reduce sequence length through compression, which might mitigate some problems associated with long contexts.
>
> ## Q3. Missing discussion on some works about adaptive tokenization, e.g.
> [1] Enhancing large language models through adaptive tokenizers NeurIPS (2024)
> [2] Task-adaptive tokenization: Enhancing long-form text generation efficacy in mental health and beyond EMNLP (2023)
>
>
> Thank you for the pointers. We actually **already** mentioned  [2] in the related work section, but [1] was indeed not previously on our radar. It appears to be a task- or domain-adaptive tokenizer training method—similar in spirit to [2], but likely more effective. We will include [1] in the related work section as a loosely related approach.
>
> That said, both [1] and [2] focus on **adaptively constructing** the tokenizer’s vocab during training or post training, which remains fixed at inference time, whereas zip2zip introduces a vocab that is **adaptive at inference time**.
>
>
> # Additional Metric&Result: Improved Perplexity via Multi-Segment Evaluation
>
>
> In light of the  suggestion from Reviewer 1 regarding segmentation ambiguity during evaluation—we implemented a **multi-view perplexity evaluation** that accounts for multiple valid LZW segmentations that decode to the same text.
>
> This refinement addresses a key challenge: unlike standard tokenizers, zip2zip can produce **multiple compressed sequences for a single target string**, complicating perplexity measurement. Our new evaluation considers alternative segmentations (pruned similarly to beam search to ensure tractability), and we find that it **substantially improves perplexity**, recovering over **50% of the gap** relative to the base model. This suggests the original drop was largely an artifact of canonical segmentation, and we're excited about the implications this has for future evaluations and model development.
>
> | **Model**  | **Method**        | **Wiki** | **Pile** | **mC4** | **dC4** |
> | ---------- | ----------------- | -------- | -------- | ------- | ------- |
> | Phi-3.5-4B | Base              | 1.62     | 1.88     | 1.94    | 1.77    |
> |            | Cont. finetune    | 1.63     | 1.89     | 1.94    | 1.77    |
> |            | zip2zip           | 1.71     | 2.02     | 2.04    | 1.84    |
> |    **NEW**       | **zip2zip multi-view** | **1.66**     | **1.92**     | **1.97**    | **1.79**    |
>
> An exmaple on how **multi-view perplexity** works
>
> #### base tokenization
> `[to, be, or, not, to, be, or, to, be, or, not]`
>
> #### canonical LZW tokenization
> `[to, be, or, not, tobe, or, tobeor, not]`
>
> #### non-canonical yet valid tokenization  (multi-view)
> - `[to, be, or, not, to, be, or, tobeor, not]`
> - `[to, be, or, not, to, be, or, tobe, or, not]`
> - `[to, be, or, not, to, be, or, to,beor, not]`
> - `[to, be, or, not, to, be, or, to,be, or, not]`
>
> Multi-view perplexity evaluation sums the probabilities across all valid tokenizations that decode to the same output. This gives a more accurate picture of how likely the model generate the ground truth text. To keep the computation tractable and avoid exponential growth in the number of sequences, we branch over **each hypertoken individually**, rather than enumerating all possible joint segmentations as an approximation.
>
> ---
>
> We hope to have been able to address all the reviewers’ concerns, are happy to answer any follow-up questions they might have, and are looking forward to their reply.

---

### Official Review · Reviewer_eQ7D · 2025-07-02

**Clarity:** 2
**Significance:** 3
**Originality:** 3
**Rating:** 5
**Confidence:** 4

**Summary:**

In this paper, they propose running LZW compression on their input, creating a codebook representing common multi-token sequences they call hyper-tokens. They then train a "hyper-token encoder" and decoder that creates an embedding representation of those hyper-tokens. These are then used as input/output of the transformer just like normal.

The hyper-token encoder/decoder is trained just like a language model, but the model should be predicting the hyper-tokens produced via compression instead of multiple steps predicting the constituent base tokens. They also include a reconstruction loss to ensure the base token embeddings can be extracted from the hyper-token.

They test the resulting models both as language models and on downstream NLP tasks. There is consistent degradation in performance, especially of language modeling, but they achieve much higher throughput due to the reduction in sequence lengths.

**Questions:**

1) Did you try using the transformers first layer as the hyper-token encoder? Works like https://arxiv.org/abs/2410.05864 suggest that LLMs already create their own aggregated representation of tokens so it could be a useful starting point. If it works so well that the hyper-token finetuning step is not needed it would really boost the impact of the work.

2) Did you try a setting where the hyper-token encoder is a small MLP with the M base token embeddings concatenated in a long vector? This is what seems to be described in the prose (section 2.2) but all the methods mentioned in the ablation in the appendix seem to be using sequence based models?

3) Did you try a setting where the hyper-token encoder is only trained on general pre-training text and then applied frozen to a new domain? The prose makes it seem like the encoder is always fine-tuned on the target domain, which runs counter to a claim about the difference between this and previous vocabulary adaptation. If the zip2zip hyper net parts could be learned once in a general way and then used on many domains while keeping a large speedup, it would really strengthen the results.

4) When looking at perplexity, did you evaluate the models just in terms of the probability it assigned to the hyper-token from LZW compression, or did you account for the probability based on other segmentations, such as using only base tokens?

**Ethical Concerns:**

["NO or VERY MINOR ethics concerns only"]

**Final Justification:**

Good work, even better with the new results in the rebuttal, but not the "groundbreaking impact" strong accept requires. Keeping score of accept.

**Limitations:**

yes

**Quality:**

3

**Strengths And Weaknesses:**

# Strengths
* The idea of the method is clearly explained and makes intuitive sense.

* Applying the compression to each input, so the first time you see a common subsequence it isn't compressed but later instances it is allows them to apply in any setting.

* Their improvements in throughput are impressive, especially on H100's

* Their idea fits nicely within the current state of how LLMs are used, their method makes minimal changes before a standard transformer is used, allowing them to apply it to many models easily

* They release fast implementations of their idea for ease-of-use

# Weaknesses
* The zip2zip introduces a lot of strange edge cases on what the model is really learning that should be discussed more. It seems like it exacerbates issues around multiple segmentations; this makes the task the LLM is learning feel very non-intuitive. For example, in figure 1, we see the first time "to be or" appears it needs to be predicted on at a time, then the next time it should predicted as a hyper token and a base token, "to be" and "or", then the third time it appears it should predicted as a single hypertoken. The fact the model needs to keep changing which representation is uses for an underlying string as it gets further into it's prediction seems like it could have large effects and should be investigated further. For example, does this large overlap in the meanings of output tokens (the hyper tokens and it's base tokens) result in systematic changes to the score distribution over tokens?

* Average reductions in sequence lengths for various datasets/domains should probably be presented somewhere in a table instead of a general "20–60% reduction" statement in the prose

* Some of the prose, especially when comparing to precious work on vocabulary adaptation makes it seem like their zip2zip works in an *inference-only* setting, i.e., that it can be applied without any training at all. Similarly, the prose is confusing w.r.t. what was actually done. Some examples include: the description of the hyper-token encoder doesn't match the models used, it isn't clear if the encoder/decoders are only trained on fire-tuning data or if there is another pre-trianing step, etc.

* Most of their results are based on their method having "relatively minor" drops in performance. This should be quantified more. How many more examples are wrong after this drop on each dataset? This is especially important given that on byte-perplexity, the one continuous task they evaluate on, their zip2zip models show consistent drops in performance on all datasets.

* Linear Scaling of codebook size. They discuss this in the limitation section, but now that some models have context lengths up to 1M tokens, this seems like a larger issue than present it as.

---

> ### Author Rebuttal · Authors · 2025-07-29
>
> We thank the reviewer for their insightful questions, which we address below and are encouraged to hear that they find the work “interesting” and “ potentially high impact".:
>
> ##  W1: Does zip2zip cause tokenization Ambiguity due to  Multi-View Segmentation ?
>
> The short answer is no. Expanded vocabulary does raise the concern of *tokenization ambiguity*—though this issue also exists in standard BPE, it’s typically less pronounced. Our use of **Lempel–Ziv–Welch (LZW) compression** offers a key advantage here: it defines a **deterministic compression rule**, where repeated sequences evolve predictably (e.g., “to”, “be”, “or” → “to be”, “or” → “to be or”) as more context accumulates. During training, the model sees exactly these patterns, and we hypothesize that it can **internalize this evolving structure**. So *in theory*, there is **no ambiguity**.
>
> In practice, the model occasionally generates suboptimal hypertokens—e.g., predicting “to be” and “or” separately instead of “to be or”—but we observe a **strong affinity toward longer, reusable hypertokens**. This behavior does introduce a challenge during evaluation: unlike standard models, where one token sequence maps cleanly to output, multiple compressed sequences can decode to the *same* text, making the space of valid predictions **exponentially large**.
>
> **(Additional Results): Improved Perplexity Under Multi-Way Tokenization**
>
> To better understand the influence of this problem, we implemented a **multi-view perplexity evaluation** that partially resolves the ambiguity. Instead of consdering only the canonical LZW segmentation, we also consider alternative segmentations that decode to the same text. (We prune some sequences—similar to beam search—to avoid exponential explosion.)
>
> The results are exciting: when accounting for multiple valid segmentations, the **effective perplexity drops significantly**, recovering **over 50% of the gap** relative to the base model. This confirms that much of the “drop” is an artifact of mismatched evaluation, and we’re thrilled by the potential this opens up!
>
> | **Model**  | **Method**        | **Wiki** | **Pile** | **mC4** | **dC4** |
> | ---------- | ----------------- | -------- | -------- | ------- | ------- |
> | Phi-3.5-4B | Base              | 1.62     | 1.88     | 1.94    | 1.77    |
> |            | Cont. finetune    | 1.63     | 1.89     | 1.94    | 1.77    |
> |            | zip2zip           | 1.71     | 2.02     | 2.04    | 1.84    |
> |    **NEW**       | **zip2zip multi-view** | **1.66**     | **1.92**     | **1.97**    | **1.79**    |
>
>
> An exmaple on how **multi-view perplexity** works
>
> #### base tokenization
> `[to, be, or, not, to, be, or, to, be, or, not]`
>
> #### canonical LZW tokenization
> `[to, be, or, not, tobe, or, tobeor, not]`
>
> #### non-canonical yet valid tokenization
> - `[to, be, or, not, to, be, or, tobeor, not]`
> - `[to, be, or, not, to, be, or, tobe, or, not]`
> - `[to, be, or, not, to, be, or, to,beor, not]`
> - `[to, be, or, not, to, be, or, to,be, or, not]`
>
> Multi-view perplexity evaluation sums the probabilities across all valid tokenizations that decode to the same output. This gives a more accurate picture of how likely the model generate the ground truth text.  To keep the computation tractable and avoid exponential growth in the number of sequences, we branch over **each hypertoken individually**, rather than enumerating all possible joint segmentations as an approximation.
>
>
>
> ## W2. Add a Dedicated Table for Sequence Length Reduction instead of an Averaging number
>
> Instead of Sequence Length Reduction, we reported Token Efficiency (bytes/token) in Table 2 because we believe it offers a more intuitive and interpretable metric. Token efficiency, measured in bytes per token, directly reflects how compactly information is encoded. The reduction in sequence length can be derived from these values, as it is inversely proportional to the efficiency gain — higher efficiency means fewer tokens are needed. We’ll clarify this relationship in the revision and add a dedicated table to explicitly report sequence length reductions. We believe including both will provide a more comprehensive view of model efficiency and thanks for the suggestion.
>
> ## W3. If zip2zip is inference-time adaptation, why there is a training step ?
>
> Zip2zip is 100% designed as **inference-time adaptation**. The reason we have a training step in the paper is not because zip2zip requires continual fine-tuning, but because we wanted to start from a **pretrained LM** (e.g., Phi or LLaMA) rather than training a model from scratch. Since these base models were not originally trained in the compressed zip2zip space, we perform a **one-time adaptation step** to enable them to operate effectively with hyper-tokens.
>
> Importantly, this is **not domain- or dataset-specific fine-tuning**—it's a one-off process. Once done, users can directly benefit from zip2zip’s **inference-time speedups** without any further training.
>
> Looking ahead, our broader vision is that future language models could be **pretrained natively with zip2zip**, enabling truly **training-free adaptive behavior out of the box**, with no additional burden on end users.
>
> ## W4. Minor drop on task performnance but more noticeable drop on perplexity ?
>
> Our response to **W1** already touches on this point. Our previous **perplexity evaluation** was unfairly penalizing **zip2zip** due to the **multi-way segmentation** issue—multiple compressed token sequences can decode to the same underlying text, yet our original evaluation only considered the single **LZW-compressed** sequence, neglecting valid alternatives.
>
> That said, many **NLP tasks**—especially **multiple-choice QA**—are robust to small changes in **language modeling quality**. These tasks often include **redundant cues** or contain clearly **incorrect distractors**, making them less sensitive to modest shifts in perplexity.
>
>
>
> ## W5. Linear Scaling of codebook size
>
> We thank the reviewer for highlighting concerns around codebook growth and inference efficiency. While it is true that the **full LZW codebook expands linearly** with context length, it is important to note that **vocabulary growth is configurable** in practice. Specifically, zip2zip supports a **maximum vocabulary size**, allowing the system to retain only the **most frequent** or **most recent hypertokens**, rather than expanding to the full set of possible entries.
>
> Moreover, even in settings where the full codebook is retained, the **runtime overhead remains modest**. Hyper-embeddings are **incrementally computed and cached** in a manner similar to standard **KV caching** in transformers, introducing only a small, constant-time cost per decoding step. We will clarify this in the revision and include additional ablation results on runtime behavior with varying vocab sizes.
>
>
> ## Q1. Using Transformer’s First Layer as Hyper-Token Encoder
>
> We thank the reviewer for pointing us to this intriguing idea and associated work. We were not previously aware that the **first transformer layer could be co-opted as a hyper-token encoder**, and we’re excited to explore this direction further.
> The approach appears promising—especially in the context of **eliminating the need for a separate hyper-encoder**. If successful, this could enable a version of zip2zip that operates **directly on top of an arbitrary pretrained LLM without requiring any adaptation phase**, which would mark an important step forward.
>
>
> ## Q2. Did you try a setting where the hyper-token encoder is a small MLP?
>
> We appreciate the suggestion and agree that using a **small MLP** as a hyper-token encoder is an interesting baseline to consider. Our current hyper-encoder is designed to be **general-purpose**, mapping a `d × M` tensor (i.e., a sequence of token embeddings) into a single `d`-dimensional hyper-token representation.. We felt MLP might be **too weak** to capture compositional patterns,  and it can't handle hyertokens with variable size, so we didn’t explore it.
>
>
> ## Q3. Did you try training the hyper-token encoder on general pretraining data and using it frozen in new domains? The prose suggests it’s always fine-tuned on the target domain.
>
> YES! All the expreiment results are using frozen pretrained model without adaptation!
>
> To clarify: the **zip2zip model**, including the **hyper-token encoder**, is trained **only once** on general pretraining data using sequences tokenized with our **LZW-based zip2zip tokenizer**.  Crucially, we do **NOT** perform any **domain-specific fine-tuning** or adaptation of the hyper-encoder. It is kept **frozen** and applied as-is across all downstream tasks and domains. This is fully consistent with our goal of supporting **inference-time adaptation**, and ensures that the same general-purpose hyper-token encoder can be deployed in new domains without retraining.
>
> We will revise the text to make this design choice more explicit. We thank the reviewer for raising this important point.
>
> ---
>
> We hope to have been able to address all the reviewers’ concerns, are happy to answer any follow-up questions they might have, and are looking forward to their reply.

---

> > ### Comment · Reviewer_eQ7D · 2025-08-08
> >
> > Thanks for the detailed response. The new multi-way perplexity numbers are encouraging for the effectiveness of zip2zip, although it does suggest that the model isn't learning the deterministic way the LZW compression works very well. Maybe future experiments on the hyper-token encoder will help learnability and put more weight on the canonical tokenization.

---

> ### Author Response · Authors · 2025-08-08
>
> Dear Reviewer eQ7D, we hope that we have addressed your questions above. As the rebuttal deadline approaches, please let us know if you still have any remaining questions. Thank you again for reviewing and the constructive feedbacks.

---

### Author Response · Authors · 2025-08-08
**General Response after Rebuttal**

We would like to thank all the reviewers for their constructive feedback and insightful suggestions. We are pleased that reviewers recognized the novelty and value of our approach. In the rebuttal phase, we added several important clarifications and results:
- (1) we introduced **multi-view perplexity evaluation**, showing that accounting for multiple valid LZW segmentations recovers over performance gap;
- (2) we clarified that **zip2zip’s adaptation step is a one-time, domain-agnostic post-training process**, after which the model dynamically adjusts its vocabulary entirely at inference time with no further training. While Reviewer-3 still holds doubt on this point, we agree to disagree.
- (3) we provided detailed discussion of **codebook growth and runtime efficiency**, explaining configurable vocabulary caps and caching strategies;
-  (4) we added results on **multilingual downstream tasks**;
- (5) we had some discussions on **quantization**, **Superword tokenizers** and **training efficiency gain**

 We believe the above additions have made our manuscript more complete and valuable for both researchers and practitioners.

---

### Note · Authors · 2025-08-11

We thank the AC and all three reviewers for their thoughtful and constructive feedback on our submission. We are encouraged by their positive assessment and the shared recognition of our work’s contributions. In particular, we appreciate the detailed and helpful comments from reviewers eQ7D and UZ7U, which have guided several improvements in our revision.


**Key clarifications during rebuttal:**

- Introduced **multi-view perplexity evaluation**, showing that accounting for multiple valid LZW segmentations recovers **over 50%** of the gap relative to the base model, clarifying that much of the drop was an evaluation artifact.
- Clarified that **zip2zip’s adaptation is a one-time, domain-agnostic post-training step**, after which the model dynamically adjusts vocabulary at inference time without further training; emphasized frozen hyper-token encoder usage across all domains.
- Expanded discussion on **codebook growth and runtime efficiency**, detailing configurable vocabulary caps, targeted merge controls (e.g., for digits), and caching strategies to sustain high throughput.
- Added **multilingual evaluations** (FR, ES, RU, ZH, AR) on TruthfulQA-2, HellaSwag, and Winograd, confirming results from main table in the paper.
- Clarified **deployment considerations**—API compatibility, modular embedding integration, batch processing, and scalability to long-sequence settings.

P.S. Although we did not receive any further response from reviewer tWZT, given that reviewer tWZT confirmed acknowledgement of our rebuttal, we interpret this as being satisfied or at least having no further doubts about our revisions.

---

### Decision · Program_Chairs · 2025-09-17

**Decision:**

Accept (poster)

**Comment:**

The paper explores an adaptive tokenization method aimed at reducing the runtime complexity of language models. The technique provided is mentioned to be original, specifically in that it achieves a speedup by modifying the tokenizer, e.g., UZ7U “This paper tackles a novel approach ... proposes adaptation to existing models that speed up inference.”,  tWZT “a high degree of originality by approaching faster inference from the perspective of tokenizer design”. In addition to this advantage related to the novelty, the impact of the method is also quite strong. The methodology is technically sound (UZ7U) and the ideas are easily applicable to many models (eQ7D). The empirical investigation is sufficiently broad (tWZT), giving convincing evidence to the generalizability of the method.

The reviewers raised a few concerns, most notably that in some cases the tokenization technique causes a drop in performance that is too significant. The authors mitigated this issue, at least partially, in their rebuttal by providing an experiment measuring “multi-view perplexity” showing that the mentioned performance drop is in fact not as severe. Given this response, the remaining concern appears to be minor. Another notable issue raised is the presentation of limitations. The rebuttal included clarifications that should be integrated into the final version.

I believe that it might take some effort, but the content of the discussion can be integrated into the paper towards a final version, and the resulting paper will be a good fit to NeurIPS